# Tropospheric and Surface Nitrogen Dioxide Changes in the Greater Toronto Area during the First Two Years of the COVID-19 Pandemic

**Xiaoyi Zhao** [1,*] , **Vitali Fioletov** [1] , **Ramina Alwarda** [1,2] , **Yushan Su** [3] , **Debora Griffin** [1] , **Dan Weaver** [4] ,
**Kimberly Strong** [2] , **Alexander Cede** [5,6] , **Thomas Hanisco** [5] , **Martin Tiefengraber** [6,7] , **Chris McLinden** [1] ,
**Henk Eskes** [8] , **Jonathan Davies** [1] , **Akira Ogyu** [1] , **Reno Sit** [1] , **Ihab Abboud** [1] and **Sum Chi Lee** [1]

1    Air Quality Research Division, Environment and Climate Change Canada, Toronto, ON M3H 5T4, Canada;
    vitali.fioletov@ec.gc.ca (V.F.); ramina.alwarda@mail.utoronto.ca (R.A.); debora.griffin@ec.gc.ca (D.G.);
    chris.mclinden@ec.gc.ca (C.M.); jonathan.davies@ec.gc.ca (J.D.); akira.ogyu@ec.gc.ca (A.O.);
    reno.sit@ec.gc.ca (R.S.); ihab.abboud@canada.ca (I.A.); sumchi.lee@ec.gc.ca (S.C.L.)
2    Department of Physics, University of Toronto, Toronto, ON M5S 1A7, Canada;
    strong@atmosp.physics.utoronto.ca
3    Ontario Ministry of the Environment, Conservation and Parks, Toronto, ON M9P 3V6, Canada;
    yushan.su@ontario.ca
4    Department of Physical and Environmental Sciences, University of Toronto at Scarborough,
    Toronto, ON M1C 1A4, Canada; dan.weaver@utoronto.ca
5    NASA Goddard Space Flight Center, Greenbelt, MD 20771, USA; alexander.cede@luftblick.at (A.C.);
    thomas.hanisco@nasa.gov (T.H.)
6    LuftBlick, Fritz-Konzert-Straße 4, 6020 Innsbruck, Austria; martin.tiefengraber@luftblick.at
7    Department of Atmospheric and Cryospheric Sciences, University of Innsbruck, 6020 Innsbruck, Austria
8    Royal Netherlands Meteorological Institute (KNMI), 3731 De Bilt, The Netherlands; henk.eskes@knmi.nl
*    Correspondence: xiaoyi.zhao@ec.gc.ca

**Abstract:** We present tropospheric nitrogen dioxide ($NO_2$) changes observed by the Canadian Pandora measurement program in the Greater Toronto Area (GTA), Canada, and compare the results with surface $NO_2$ concentrations measured via in situ instruments to assess the local emission changes during the first two years of the COVID-19 pandemic. In the City of Toronto, the first lockdown period started on 15 March 2020, and continued until 24 June 2020. ECMWF Reanalysis v5 (ERA-5) wind information was used to facilitate the data analysis and reveal detailed local emission changes from different areas of the City of Toronto. Evaluating seven years of Pandora observations, a clear $NO_2$ reduction was found, especially from the more polluted downtown Toronto and airport areas (e.g., declined by 35% to 40% in 2020 compared to the 5-year mean value from these areas) during the first two years of the pandemic. Compared to the sharp decline in $NO_2$ emissions in 2020, the atmospheric $NO_2$ levels in 2021 started to recover, but are still below the mean values in pre-pandemic time. For some sites, the pre-pandemic $NO_2$ local morning rush hour peak has still not returned in 2021, indicating a change in local traffic and commuter patterns. The long-term (12 years) surface air quality record shows a statistically significant decline in $NO_2$ with and without April to September 2020 observations (trend of $-4.1\%$/yr and $-3.9\%$/yr, respectively). Even considering this long-term negative trend in $NO_2$, the observed $NO_2$ reduction (from both Pandora and in situ) in the early stage of the pandemic is still statistically significant. By implementing the new wind-based validation method, the high-resolution satellite instrument (TROPOMI) can also capture the local $NO_2$ emission pattern changes to a good level of agreement with the ground-based observations. The bias between ground-based and satellite observations during the pandemic was found to have a positive shift (5–12%) than the bias during the pre-pandemic period.

**Keywords:** $NO_2$; Pandora; COVID-19

## 1. Introduction

Tropospheric nitrogen dioxide (NO$_2$), an important primary air pollutant, has negative health and environmental impacts [1,2]. Primary atmospheric sources of tropospheric NO$_2$ pollution are fossil fuel combustion and biomass burning. Together with surface ozone and fine particulate matter (PM$_{2.5}$), NO$_2$ is one of the three major air pollutants that have been used to calculate the Canadian Air Quality Health Index (AQHI) [3]. Exposure to NO$_2$ can lead to decreasing lung function and increases the susceptibility to allergens for people with asthma [4,5]. Excessive deposition of NO$_2$ to ecosystems can lead to acidification of soil and water and, ultimately, critical load exceedance [6].

Traditionally, NO$_2$ atmospheric monitoring is completed via ground-level in situ instruments. In Canada, surface air quality monitoring in populated regions is carried out by the National Air Pollution Surveillance (NAPS) program [7]. In situ observations can provide continuous air quality monitoring with good accuracy. However, in situ instruments are mostly sensitive to pollutants emitted at the surface and cannot monitor vertical profile and elevated (above-ground) transport of pollutants. To address this limitation, ground-based UV-visible remote sensing instruments able to operate in multiple observation modes (direct-sun, zenith-sky and off-axis spectroscopy techniques) are becoming increasingly common. These provide information on the column density of NO$_2$ as well as some limited vertical profiles [8–11]. More advanced algorithms can be used to further estimate surface NO$_2$ [12–14]. Among the more recent examples of deployed UV-visible remote sensing instruments, the Pandora sun spectrometer has proven to deliver high quality and good precision NO$_2$ observation data products [15–18]. The Pandora observation program was initiated by the National Aeronautics and Space Administration (NASA) in 2006. In 2019, it evolved (due to funding from the European Space Agency (ESA) from 2014 on) to the current Pandonia Global Network (PGN), where the PGN is an international global collaboration between NASA, ESA and the US Environmental Protection Agency (US-EPA) [19]. Environment and Climate Change Canada (ECCC) has been one of the early research partners with the NASA Pandora project since 2013 [19,20]. After almost one decade since its first deployment, ECCC currently operates ten Pandora instruments at eight Canadian sites to perform satellite data validation and interpretation, air quality monitoring, polar stratospheric ozone depletion studies, and remote sensing technique research (e.g., [12,17,21,22]). A dedicated high-density Canadian Pandora observation sub-network has been established, which, as of 2021, consists of five sites in and near the Greater Toronto Area (GTA) with one major goal being the validation of higher resolution satellite instruments (e.g., TROPOMI and TEMPO instruments [21,22]). In this work, Pandora observations from Downsview, the University of Toronto St. George campus (UTSG), and the University of Toronto Scarborough campus (UTSC) sites have been used to analyze and illustrate the NO$_2$ pollution changes during the first two years of the SARS-CoV-2 (i.e., COVID-19) pandemic (i.e., 2020 and 2021).

Since 2020, many research studies have been carried out to evaluate the NO$_2$ pollution changes during the early period or first year of the COVID-19 pandemic. Numerous studies were completed regionally (e.g., [23–29]) and globally (e.g., [30–32]), with various observations' methods. For example, Bauwens et al. [30] reported the average NO$_2$ column, observed by TROPOMI, dropped by 40%, 38%, and 20%, over Chinese, American, and European cities, respectively, by April 2020 compared to pre-COVID pandemic measurements. Cooper et al. [31] quantified NO$_2$ changes in more than 200 cities worldwide and reported mean surface NO$_2$ concentrations are 29% $\pm$ 3% lower in countries with strict lockdown conditions than in those without in 2020. To our best knowledge, this paper is the first peer-review study of the COVID-19 pandemic NO$_2$ changes that cover the period of not only 2020 but also 2021.

In addition to the Pandoras, long-term surface in situ observations of NO$_2$ from the NAPS program have also been analyzed to reveal the general NO$_2$ trends in Toronto. In addition, total column NO$_2$ data products (the standard ESA data product using the NO$_2$ retrieval algorithm developed by Royal Netherlands Meteorological Institute (KNMI) [33]

and a research data product developed by ECCC [18]) from the TROPOMI satellite instrument are used to further assess the $NO_2$ emission changes. The ECMWF Reanalysis v5 (ERA-5) wind data were merged with remote sensing and in situ datasets to perform (1) wind-rotation satellite validation; and (2) regional air pollution sources analysis [18].

The paper is organized as follows: Section 2 describes the remote sensing and in situ sites and datasets that have been used in this work. Section 3 shows the analysis results for ground-based remote sensing instruments (Pandora sun spectrometers), which are the regional $NO_2$ emission changes due to the pandemic. In Section 4, surface in situ observations are presented and used to calculate the general air quality trends in the area. In Section 5, KNMI (S5P-PAL reprocessed; Sentinel-5P Product Algorithm Laboratory) and ECCC TROPOMI satellite data products are presented to evaluate their performance and agreement with ground-based observations made during the decreased $NO_2$ emissions in the area. Lastly, Section 6 gives the conclusions.

## 2. Observation Sites and Datasets

### 2.1. Observation Sites

The GTA Pandora sites are selected to be close to the NAPS' surface air quality monitoring stations and are also geographically distributed to represent different parts of the City of Toronto. A map of the Pandora and NAPS in situ sites is shown in Figure 1.

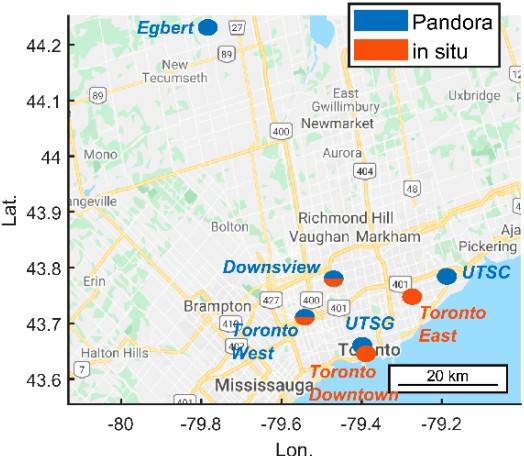

**Figure 1.** Pandora and in situ observation sites in and near the Greater Toronto Area (base map from © Google Maps).

Pandora observations at the Downsview site (43.781° N, −79.468° W, 187 m a.s.l.) started in 2013. However, due to the entrance window not being optimized for $NO_2$ observations, the high-quality $NO_2$ data products only started in 2015. The second Pandora site was at the University of Toronto St. George campus (UTSG; 43.661° N, −79.399° W, 176 m a.s.l.) in 2016. The Egbert site, equipped with a Pandora instrument in 2018, is the background air quality monitoring site in a rural area (north of the city; 44.230° N, −79.780° W, 251 m a.s.l.). Another two Pandora instruments were deployed at the University of Toronto Scarborough campus (UTSC; 43.784° N, −79.187° W, 137 m a.s.l.) and NAPS Toronto West (43.709° N, −79.544° W, 141 m a.s.l.) sites in 2019 and 2021, respectively. In this work, we will only present Pandora observations from Downsview, UTSG, and UTSC sites. Pandora observations from the Egbert and Toronto West sites were excluded in this work because the instrument at Egbert had a technical issue in 2020, with a new instrument deployed in 2021, and the Toronto West site has only a short period of observations.

Four NAPS in situ sites (Toronto North, Toronto Downtown, Toronto West, and Toronto East) are close to or collocated with Pandora sites in Toronto (Figure 1, see red symbols). These NAPS sites have had continuous observations for more than a decade, although two sites were relocated since 2017. The Toronto North site was relocated from Hendon Ave. and Yonge St. (43.782° N, −79.418° W) to Downsview in January 2017.

Although the site has only been relocated by about 4 km, some of the local $NO_2$ pollution patterns are different (e.g., the direction of major pollution sources). The NAPS Toronto Downtown site was relocated from Bay St. and Wellesley St. W. ($43.663°$ N, $−79.388°$ W) to Metro Hall ($43.645°$ N, $−79.389°$ W) in June 2019. Thus, extra caution was taken when analyzing trends in these in situ data records (see Appendix A).

### 2.2. Pandora Sun Spectrometer

The 1-S type Pandora instrument measures direct-sun (DS) and scattered sunlight in the UV-visible spectral range (280–530 nm) with a spectral resolution of 0.6 nm [15]. In this work, total column $NO_2$ (integrated $NO_2$ amount from the surface to the top of the atmosphere) is produced using Pandora's standard $NO_2$ algorithm with the total optical absorption spectroscopy (TOAS) technique [13]. Zhao et al. [18] demonstrated that the Pandora DS $NO_2$ column data have a precision better than 0.02 DU (Dobson Unit; 1 DU = $2.6870 \times 10^{16}$ molec $cm^{−2}$). In this work, $NO_2$ total column data were used from Pandora nos. 103, 109, and 145 at the Downsview (where the current NAPS Toronto North site was relocated to in 2017), UTSG, and UTSC sites, respectively.

To better isolate the $NO_2$ pollution from potential changes due to variation of "background" stratospheric $NO_2$, tropospheric $NO_2$ column values were generated for these instruments by subtracting estimated stratospheric $NO_2$ values as:

$$VCD_{trop}(t) = VCD_P(t) − SVCD_{OMI}(t_0) \times R(t, t_0) \tag{1}$$

where, $VCD_{trop}$ is the tropospheric vertical column density of $NO_2$ at observation time $t$, $VCD_P$ is the total vertical column $NO_2$ observed by Pandora, $SVCD_{OMI}(t_0)$ is the stratospheric $NO_2$ amount observed by Ozone Monitoring Instrument (OMI) (SPv3) [34] at its overpass time $t_0$. $R(t)$ is the diurnal conversion factor calculated using the PRATMO stratospheric photochemical box model [35,36] that adjusts OMI measured stratospheric $NO_2$ at $t_0$ to Pandora observation time $t$ [12].

### 2.3. TROPOMI

The TROPOMI instrument on board the Sentinel 5 Precursor satellite measures a wide spectral range with its UVN module (Ultraviolet to near-infrared). The standard $NO_2$ retrieval products are produced with an improved DOMINO retrieval algorithm [33,37]. The overall uncertainty of TROPOMI data is about 0.032 DU.

The most recent data product, processor version 2.3.1, was used here. This dataset became available in December 2021 on the S5P-PAL data portal (https://data-portal.s5p-pal.com/, last accessed on 21 February 2022). It provides consistent reprocessing from May 2018 to November 2021, connecting to the official offline product for later dates [33].

Data were required to have a quality flag qa > 0.75 and cloud fraction < 0.3. Using the recalculated high-resolution tropospheric air mass factor (AMF), ECCC developed and validated a high-quality research $NO_2$ product for TROPOMI [17,18,38]. Benefitting from the high-resolution AMFs (improved from 40 km $\times$ 110 km to 10 km $\times$ 10 km in the GTA), the ECCC recalculated $NO_2$ product has improved agreement with ground-based observations (e.g., lower bias and improved precision) [18]. Detailed uncertainties' estimations of TROPOMI $NO_2$ data products can be found in Zhao et al. [18]. As a sun-synchronous satellite instrument, TROPOMI only measures once per day over most mid-latitude regions (including the GTA). At nadir, TROPOMI pixel sizes were 3.5 $\times$ 7 $km^2$ at the beginning of operation and were reduced to 3.5 $\times$ 5.6 $km^2$ on 6 August 2019.

In this work, we implemented the newly developed and validated wind-rotation-based satellite validation technique [18] instead of the traditional satellite validation method (i.e., simple pair satellite overpass pixel with ground-based observation at the same time). It was proven that the new method could increase the number of coincident measurements between ground-based and satellite instruments by a factor of five [18].

### 2.4. In Situ

Surface air quality data were used from the NAPS monitoring sites located in Toronto (see Figure 1). The surface $NO_2$ concentration was measured by Thermo Scientific Model 42i Analyzer with a lower detectable limit of 0.4 ppbv and precision of 0.4 ppbv. In this work, quality-controlled hourly averaged data were used. The long-term $NO_2$ concentration data have been used to investigate if the pandemic significantly impacted the $NO_2$ trend at the observation site. More details about the trend analysis are provided in Section 4.2.

### 2.5. ERA-5

ECMWF Reanalysis v5 (ERA-5) wind data were used to study regional air quality conditions, i.e., to understand the direction of pollution sources and to isolate meteorological factors (e.g., wind directions) that can affect the regional air quality. Remote sensing (satellite and ground-based) and in situ observations were merged with averaged surface layer winds (i.e., from 1000 to 900 hPa) [18]. More details about the ERA-5 wind data and wind averaging can be found in Zhao et al. [18].

## 3. Ground-Based Remote Sensing Observations

### 3.1. Downsview

Figure 2 shows the monthly mean tropospheric $NO_2$ column time series measured at the Downsview site for different years. The record shows tropospheric $NO_2$ had a clear decline in 2020 from the previous five-year mean (i.e., the red line versus the black dashed line with the 1-sigma envelope). The monthly data show the $NO_2$ decline started in April 2020 (in the City of Toronto, the first lockdown period started on 15 March 2020, and continued until 24 June 2020) and returned to be within the 1-sigma envelope in November 2020. For most of 2021, the $NO_2$ level remains at the lower bottom of the five-year 1-sigma envelope.

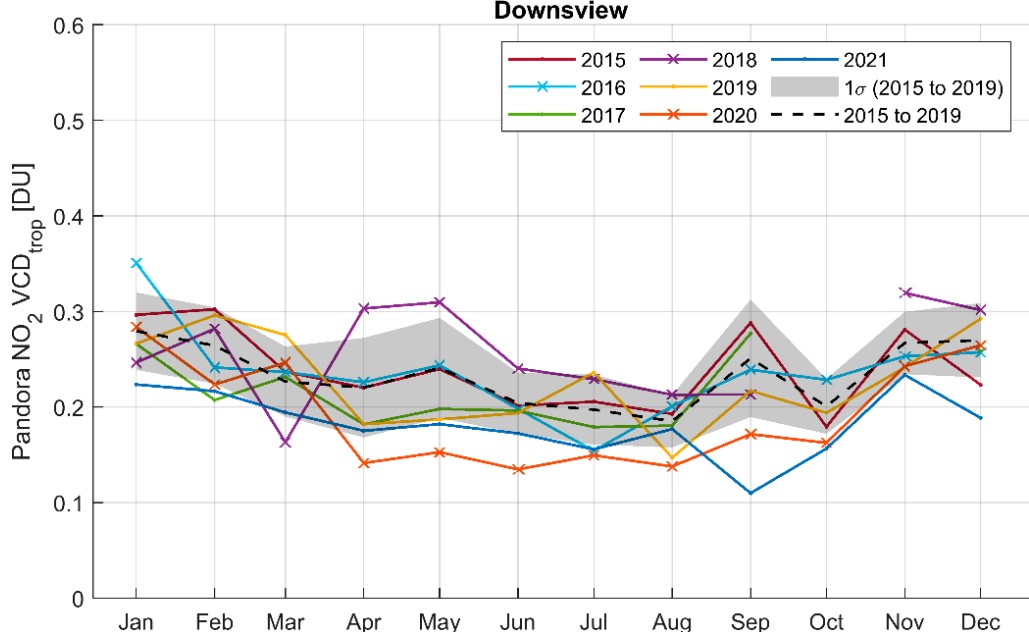

**Figure 2.** Time series of monthly $NO_2$ tropospheric column at Downsview site. The black dashed line is the mean of observations from 2015 to 2019, with the shading area representing the standard deviation of the 5-year mean.

The City of Toronto and the Province of Ontario had several rounds of lockdown and stay-at-home orders, which clearly affected the observed $NO_2$ in this area during different pandemic periods. Thus, to better understand and quantify the impact of the pandemic, we performed an additional analysis with a focus on observations from 15 March

to 15 September of each year. This 6-month period covers the period in 2020 that shows the most significant decreased $NO_2$ level. Figure 3 shows the histograms of the observed tropospheric $NO_2$ categorized into weekday and weekend. Before the pandemic, the site typically observes 0.20 DU and 0.13 DU tropospheric $NO_2$ during weekdays and weekends, respectively. However, since the pandemic, these numbers decreased to 0.14 DU and 0.1 DU, respectively. There was a more pronounced decrease for weekdays, with more visible changes in the distribution of the dataset. It is worth noting that, during the pandemic, the tropospheric $NO_2$ levels during the weekdays were almost similar to what had been observed during the weekends in pre-pandemic times (see Figure 3b,c).

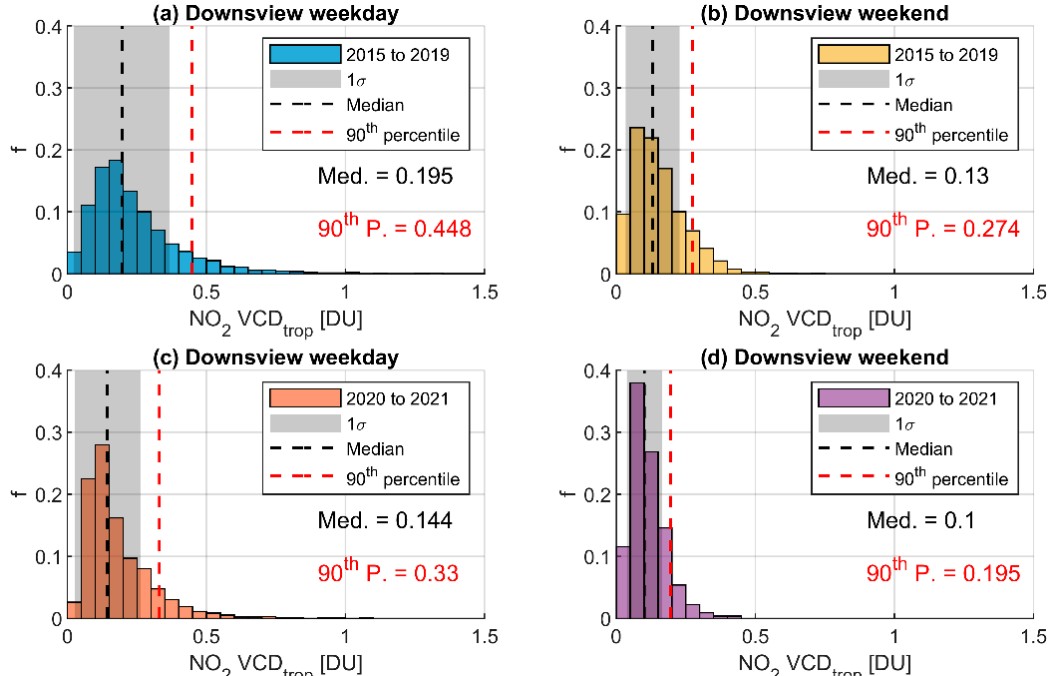

**Figure 3.** Histograms of $NO_2$ tropospheric column at Downsview site for observations from 15 March to 15 September for each year. (**a**,**b**) show weekday and weekend observations before the pandemic (2015–2019), respectively; (**c**,**d**) show weekday and weekend observations during the pandemic (2020–2021), respectively. The vertical black dashed line shows the median value of tropospheric $NO_2$ with shading area representing its standard deviation. The vertical red dashed line shows the 90th percentile of observed tropospheric $NO_2$.

However, before we can draw any conclusions about the cause of decreasing tropospheric $NO_2$ at this site, we must consider if the meteorological conditions were different. The Downsview site is located on the northern edge of the City of Toronto. Thus, most of the observed pollutants came from downtown, or from the south. Figure 4a shows the observed tropospheric $NO_2$ binned by wind directions. The $NO_2$ decrease is not homogeneous, but depends on the direction of the winds. For the polluted directions (i.e., around 180 to 240 degrees), a clearer decline of $NO_2$ emissions can be observed (i.e., decreased by more than 0.1 DU). For example, observations from 180° and 210° directions (downtown and airport directions) declined by 40% and 35% in 2020 compared to the 5-year mean value, respectively. In contrast, when observed air was from less polluted directions (i.e., north of the site, from 330 to 30 degrees), the decline is much weaker (less than 0.05 DU; half of the decreased amount from polluted directions). These results indicate that pollution reduction is not uniform across the entire city and suggests decreased traffic in the downtown areas and the decreased air traffic from city airports contributed more than other $NO_2$ emission sectors (e.g., local residential emissions). This is consistent with a modelling study [39] in which it was estimated that the pandemic led to a 60% decrease in on-road emissions and 80% in airport landing and takeoffs during the most intensive phase of the lockdown in the

GTA. It is also consistent with reductions in CO and $CO_2$ measured with open-path Fourier transform spectroscopy at UTSG and attributed to reduced traffic during the COVID-19 lockdown [40].

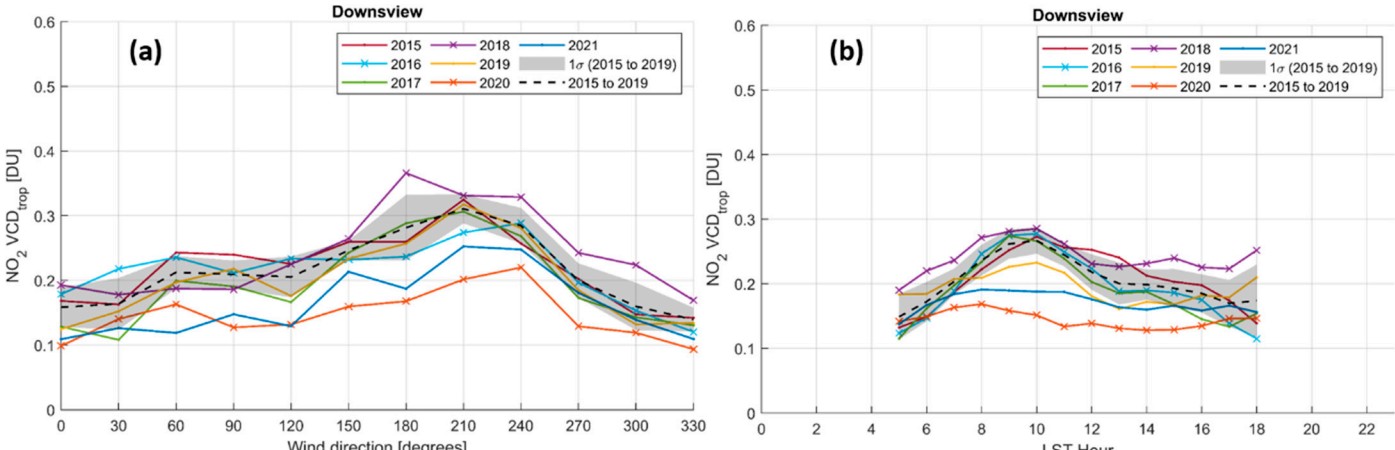

**Figure 4.** $NO_2$ tropospheric column at Downsview site binned by (**a**) wind directions; and (**b**) local standard time (LST) in hours for observations from 15 March to 15 September of each year. The black dashed line is the mean of observations from 2015 to 2019, with the shading area representing the standard deviation of the 5-year mean.

The $NO_2$ reduction in 2020 and 2021 is nonhomogeneous, not only spatially but also temporally. As shown in Figure 4b, a clear morning rush hour emission peak (around 9 to 10 of the clock; local standard time, LST) is prominent pre-pandemic. However, this local traffic signal disappeared in 2020 and 2021. For example, compared to the 5-year mean, the $NO_2$ levels at 10 am decreased by 43% and 30% in 2020 and 2021, respectively. Note that, the cause of this peak is not only due to traffic emissions but also boundary layer dynamics (e.g., [38]; see more discussions in Section 4). Compared to 2020, the overall $NO_2$ level increased by about 0.02 to 0.05 DU from 8 am to 11 am in 2021, which indicates a recovery of local traffic. However, the rush hour emission peak was still missing in 2021. These results reflect the changing travel and local commuting behavior due to the pandemic.

### 3.2. UTSG and UTSC

The second-longest Pandora timeseries is at the UTSG site (five years), which is located in downtown Toronto. As shown in Figure 5, the $NO_2$ pollution decreased during the pandemic period for weekdays (median value decreased from 0.22 DU to 0.18 DU), but has almost no changes for weekends. However, similar to the Downsview site, this reduction is not homogeneous. Air masses coming from south to south-west had the most reduction in $NO_2$ (see Figure 6c). In addition, there was also a reduction in the morning rush hour peak (Figure 6a); the rush hour peak was reduced from 0.31 DU to 0.25 DU at 9 am. The continued presence of the rush hour peak at this downtown site in 2020 and 2021 is in contrast to that of the Downsview site.

Although the UTSC site only had Pandora observations during the pandemic (from 2020 to 2021), it shows that 2021 $NO_2$ is higher than in 2020, as shown in Figure 6b. Figure 6a,b show that the overall pollution level at UTSC is much lower than UTSG during the rush hours (i.e., the highest values for UTSG and UTSC in 2020 are 0.25 DU and 0.14 DU, respectively). However, the difference became smaller in the afternoon. Not surprisingly, for the UTSC site, the high $NO_2$ pollution is from downtown Toronto (210° to 300° degrees; see Figure 6d). In short, these remote sensing observations reveal that, even within the city (Downsview and UTSC are 14 km and 22 km away, from UTSG/downtown, respectively), the local air pollution level could be very different throughout the day (e.g., controlled by boundary layer dynamics and local traffic conditions) and highly depends on local meteorological conditions.

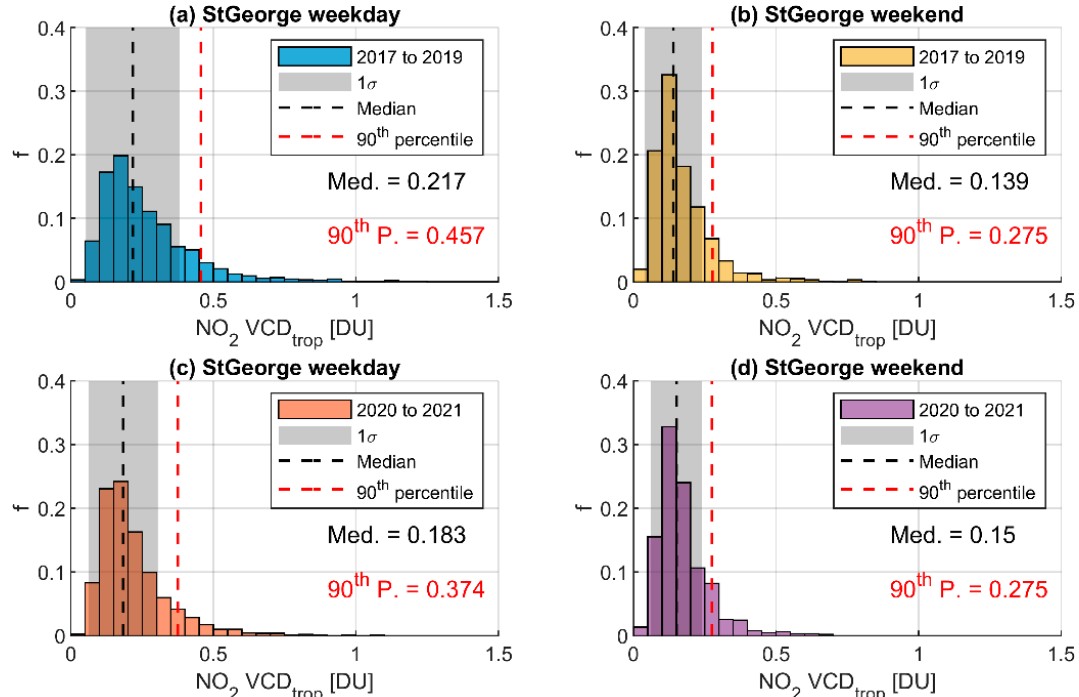

**Figure 5.** Histograms of $NO_2$ tropospheric column at UTSG site for observations from 15 March to 15 September of each year. (**a**,**b**) show weekday and weekend observations before the pandemic (2017–2019), respectively; (**c**,**d**) show weekday and weekend observations during the pandemic (2020–2021), respectively. The vertical black dashed line shows the median value of tropospheric $NO_2$ with shading area representing its standard deviation. The vertical red dashed line shows the 90th percentile of observed tropospheric $NO_2$.

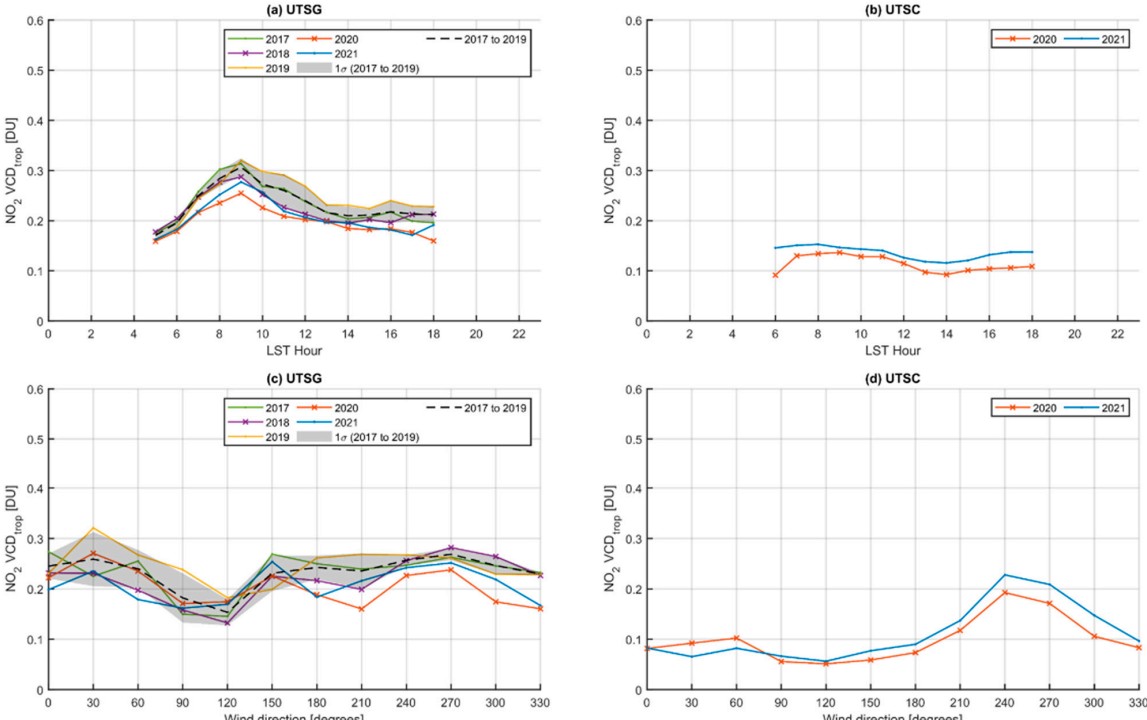

**Figure 6.** Pandora measured $NO_2$ tropospheric column at UTSG (**a**,**c**) and UTSC (**b**,**d**) sites binned by local standard time (LST) in hours and wind directions (observations from 15 March to 15 September of each year). The black dashed line is the mean of observations from 2017 to 2019, with the shading area representing the standard deviation of the 5-year mean.

## 4. In Situ Records

### 4.1. Temporal and Spatial Distributions

Several studies utilizing in situ observations in GTA show the regional air quality changes due to the COVID-19 pandemic (e.g., [41,42]). As described in Section 2.1, the Toronto North site (in situ) was relocated to Downsview in 2017. The relocation of 4 km led to different patterns of surface $NO_2$ (see Appendix A) as related to wind direction. Thus, Figure 7 only shows results from in situ observations from the new NAPS Toronto North site at Downsview (co-located with Pandora) and Toronto East site (close to UTSC).

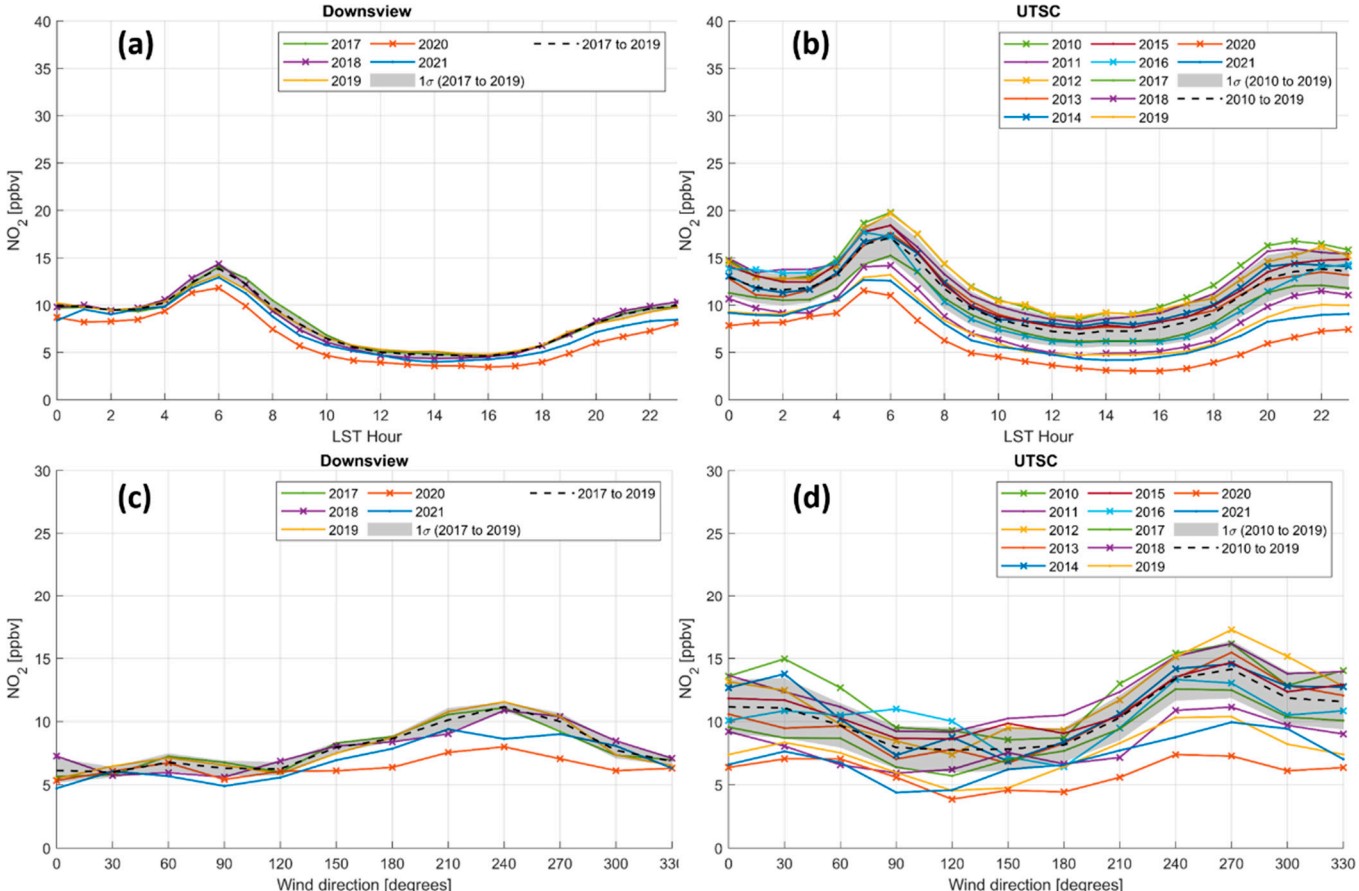

**Figure 7.** In situ measured $NO_2$ surface concentration at Downsview (**a**,**c**) and Toronto East (**b**,**d**) sites binned by local standard time (LST) in hours and wind directions (observations from 15 March to 15 September for each year). The black dashed line is the mean of observations from pre-pandemic with shading area representing the standard deviation of the mean.

Both sites show enhanced morning surface $NO_2$ around 6 am, even in 2020 (see Figure 7a,b). Similar to observations from the Pandoras, in situ data reveal that the $NO_2$ reduction in 2020 and 2021 is nonhomogeneous, not only spatially but also temporally (i.e., more reduction from downtown directions and more reduction during the morning and evening rush hours). The diurnal pattern of surface $NO_2$ (Figure 7a,c) looks similar but not identical to the results of tropospheric column $NO_2$ (i.e., see Figures 4b and 6b, which show peaks at 9 am). However, these results are not inconsistent with the observations from the Pandora instruments. Note that, theoretically, the Pandora instruments' sample $NO_2$ molecules within the entire tropospheric column, while in situ instruments only observe $NO_2$ at the surface level. In the early morning, with relatively lower temperature on the surface and less boundary layer dynamics, $NO_2$ emissions are more likely abundant near the surface than distributed and diluted vertically in the later hours (e.g., [43,44]). Thus, it is also worth noting that the amplitude of diurnal variations for surface concentrations is

more than twice that of Pandora's tropospheric $NO_2$ measurements, as expected. Another factor that should be considered is the altitude differences between Pandora and in situ instrument. The in situ instruments are mostly located on road levels, while Pandoras are located on rooftops (to have better viewing conditions). These differences would further complicate the comparison.

On the other hand, when binned by wind directions (Figure 7c,d), in situ data show good agreement with the Pandora data (see Figures 4a and 6d). Both in situ and Pandora data show peaks for the same wind directions (i.e., downtown Toronto). In short, these results demonstrated that Pandora instruments could be used to augment local air quality monitoring and provide valuable information about pollutants above ground level.

### 4.2. Surface $NO_2$ Trends

Surface measurements in Toronto demonstrate a long-term decline in $NO_2$ concentrations that should be taken into account when we compare 2020 and 2021 data with the observations from previous years. Surface $NO_2$ has a strong seasonal cycle and must be treated carefully to reveal trends and uncertainties. For example, Mashayekhi et al. [45] identified a higher seasonal decline in 2020 and show that, by subtracting the seasonal changes in 2020, the COVID-19-induced $NO_2$ surface concentrations reduction is 6% for Toronto for the lockdown period (without considering meteorological variations). Following the development of statistical models that account for the changing amplitude of the seasonal cycle previously [46–48], the surface $NO_2$ data have been decomposed into linear and changing amplitude seasonal components.

$$\Omega = \beta t + \alpha(t) + \sum_{i=1}^{2} I\left( b_{1i} \sin\left(\frac{2i\pi t}{12}\right) + b_{2i} \cos\left(\frac{2i\pi t}{12}\right) \right) + N \tag{2}$$

Here, $\Omega$ is the $NO_2$ amount; $t$ represents time (in months); $\beta t$ is the linear trend term; $\alpha$ is time-dependent seasonal offset; the summation operator term is the seasonal signal with a local regression indicator; $b_{1i}$ and $b_{2i}$ are the constant coefficients to be determined from the fitting; and $N$ is the residual. The local regression indicator fits the data with a time window of $\pm 6$ months of each year, e.g., for the year 2015, the seasonal signal will be fitted with data from July 2014 to June 2016. The uncertainty of the trend is calculated following Weatherhead et al. [49].

As illustrated in Section 3 (e.g., Figure 2), the most prominent reduction of $NO_2$ was observed from April to September 2020. To understand the impact of the pandemic, the data were fitted for (1) the entire records and (2) the records without April to September 2020, separately. Figure 8 shows the statistical model fitted results for the NAPS Toronto East site.

Figure 8 shows that, with or without April to September 2020 data, the surface $NO_2$ always has a significant decreasing trend (95% confidence level). Expectedly, when including observations from April to September 2020, the amplitude of the decreasing trend became slightly larger (i.e., changing from $-0.57 \pm 0.04$ ppbv/yr to $-0.60 \pm 0.04$ ppbv/yr). Both trends are significant, but the observations from April to September 2020 are visible outliers (see red dots for this period in Figure 8b).

Figure 8b shows that the de-seasonal data during this early stage of the pandemic are all below the fitted trend line (i.e., all red dots from April to September 2020 are below the fitted black dashed line), with the mean value of 1.5 ppbv lower than the fitted trend (which represents the expected decline, i.e., without the pandemic-induced decline during the early stage of the pandemic in 2020). On the other hand, the standard deviation of the de-seasonal data (March to September of each year, but without 2020 data) is only 1.3 ppbv and therefore the standard error of a six-month mean is 0.53 ppbv. Thus, we conclude that even considering the overall $NO_2$ decreasing trend in this area, the surface $NO_2$ reduction in this early pandemic period is still significant on more than the 2-sigma level.

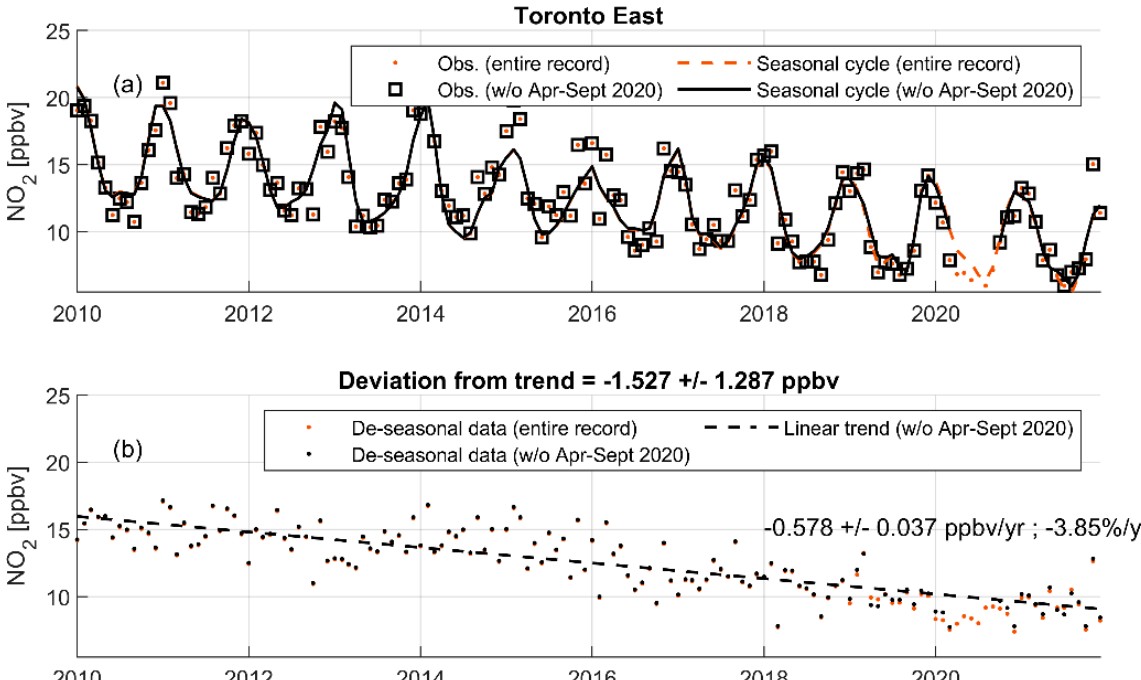

**Figure 8.** Surface $NO_2$ trend fitting results from the statistical model at the Toronto East site. Red symbols represent data and fittings for the entire observation record. Black symbols represent data and fittings for the records without data from April to September 2020. Panel (**a**) shows the observations and fitted seasonal cycles; panel (**b**) shows the de-seasonal data and fitted linear trend.

Note that if the surface data had been fitted with only pre-pandemic data, the trend was $-0.56 \pm 0.04$ ppbv/yr ($-3.75\%$/yr). Thus, unlike the significant decrease in the early stage of the pandemic, although the $NO_2$ conditions in 2021 are still affected by the COVID-19 pandemic as illustrated in previous sections, it is not significant when considering the decreasing trend.

## 5. Comparison with Satellite Observations

To study the impact of the COVID-19 lockdown in Toronto using TROPOMI satellite data (version 2.3.1), satellite measurements were first compared to the Pandora observations. The standard approach is to find coincident ground-based and satellite observations with certain criteria (e.g., temporal, spatial, and quality control criteria). For example, in our previous work [18], the coincident criteria used to pair Pandora and TROPOMI were selected as (1) the nearest (in time) measurement that was within $\pm 10$ min of the TROPOMI overpass time; (2) the closest TROPOMI ground pixel (having a distance of less than 10 km from the ground pixel center to the location of the Pandora instrument); and (3) TROPOMI $NO_2$ data product quality flag > 0.75 [50]. For satellites typically with one overpass per day, only one coincident data point could be acquired per day, assuming optimal conditions (weather and instrumental). Thus, to fully utilize the modern high-resolution satellite, we developed and validated a new wind-based validation method [18], which could increase the coincident data number by a factor of five. The larger number of coincident measurements can greatly improve the statistics of the dataset and provide a better understanding of the spatial and transport pattern of $NO_2$. Thus, in this work, the wind-based method is used for the three Pandora sites following the data selection and filter criteria described in Zhao et al. [18]. As this work focuses on understanding the impact of the pandemic on local $NO_2$ emission, we only use observations from 15 March to 15 September of each year.

Utilizing the pixel-averaging technique [51,52], Figure 9 shows the TROPOMI ECCC recalculated tropospheric $NO_2$ columns averaged over the defined pre-pandemic and

pandemic periods. Figure 9c,d shows that the difference between the two periods is visible and mostly prominent in the downtown areas (decreased by about $-0.04$ DU or $-25\%$). Griffin et al. [39] showed that TROPOMI observed $NO_2$ decreases in parts of the GTA can even exceed $-60\%$ during the early stage of the pandemic (i.e., 16 March to 8 May 2020).

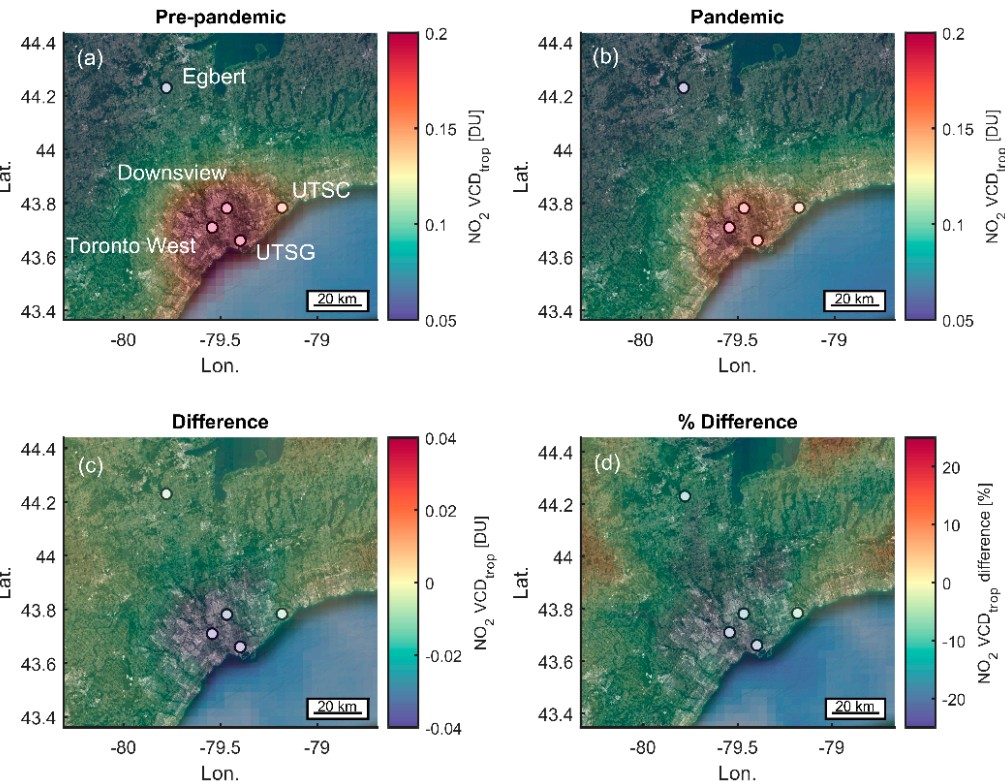

**Figure 9.** TROPOMI ECCC recalculated tropospheric $NO_2$ columns smoothed by pixel averaging (with observations from 15 March to 15 September for each year). Pandora sites show as white dots. (**a**) data before COVID-19 pandemic (i.e., 2018–2019; note 2018 data start since May); (**b**) data during the pandemic (i.e., 2020–2021); (**c**) difference between the two periods; (**d**) percentage difference between the two periods. Base map from Google Maps©.

Figure 10 shows the regression analysis for TROPOMI- and Pandora-observed $NO_2$ total columns. Here, only the ECCC re-calculated $NO_2$ product is used (TROPOMI S5P-PAL reprocessed product results are provided in Appendix B; note that the ECCC total column $NO_2$ was calculated as ECCC recalculated tropospheric $NO_2$ + S5P-PAL stratospheric $NO_2$, more details can be found in [18]). Following the previous analysis method for Pandora and in situ data, here the analysis only uses data from March to September of each year. The comparison of results for the pre-pandemic and pandemic periods demonstrates similar systematic differences between TROPOMI and Pandora data: a better agreement between satellite and ground-based instruments at Downsview and UTSC sites than at UTSG (downtown Toronto). The largest bias is from the UTSG site that is $-17\% \pm 1.2\%$ and $-12\% \pm 0.7\%$ for pre-pandemic and pandemic periods, respectively. This is in line with the previous TROPOMI validation studies (e.g., [18,53]; note these studies were related to v1 of TROPOMI data) that demonstrate that TROPOMI tends to underestimate tropospheric $NO_2$ in heavily polluted areas (e.g., due to limited resolution [54]). Based on current results, the latest S5P-PAL has less difference compared with the ECCC recalculated products than previous versions [18].

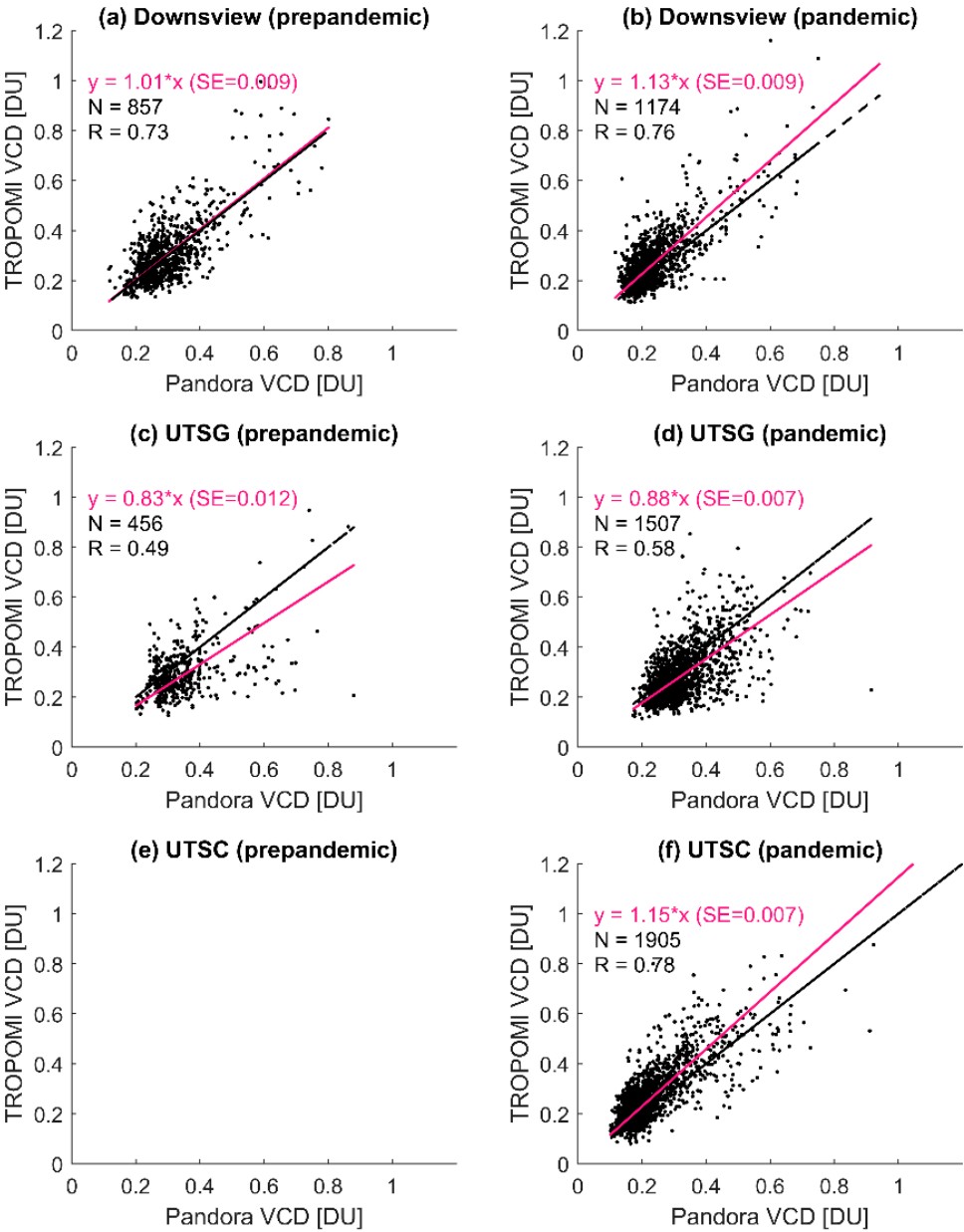

**Figure 10.** TROPOMI vs. Pandora $NO_2$ total column (VCD) measurements at Downsview, UTSG, and UTSC, using data from before the pandemic (**a**,**c**,**e**) and since pandemic (**b**,**d**,**f**). The magenta line is the linear fit with intercept set to 0 (fitting result and standard error are shown on each plot), and the black line is the one-to-one line. Pandora measurements taken within ±1 h around the TROPOMI overpass time were used (observations from 15 March to 15 September for each year).

For Downsview, the bias was increased in magnitude during the pandemic (i.e., from 1% ± 0.9% to 13% ± 0.9%). The UTSC site had no corresponding observations before the pandemic (e.g., March to September in 2019), but it shows good results when compared with TROPOMI with 15% ± 0.7% positive bias and a correlation coefficient of 0.8. The results indicate there might be a positive shift (5–12%) in the bias between ground-based and satellite observations due to the large change of emissions in city areas. The cause of this shifting bias could be due to (1) challenges in producing reliable a priori for the pandemic periods (i.e., large changes in local emissions conditions) and (2) pixel size changes of TROPOMI $NO_2$ data products (note that since 6 August 2019, the resolution has improved from 5 km × 5 km to 3 km × 5 km). Please note that based on TROPOMI's resolution, the theoretical bias between satellite and ground-based observations is about

−10% [17]. More research and validation work is needed to fully understand these changes. However, to verify this, similar analyses from more ground-based sites should be included.

For Downsview and UTSG, reduction in NO₂ total columns can be found during the pandemic and is consistent with the findings from Section 3 (e.g., Pandora data show a 0.05 DU reduction at Downsview). TROPOMI shows a lower reduction and a clear systematic bias that can be found for the UTSG site (see Figure A6). As illustrated in Sections 3 and 4, the NO₂ reduction is not homogeneous in space and time. Thus, the coincident data were binned by wind directions to reveal whether TROPOMI also captured the regional emission changes. Figure 11 shows that the satellite data tracked the ground-based observation pattern very well for these two periods. Griffin et al. [39] also shows that in 2020, the largest NO₂ emission change in southern Ontario was from the reduction of traffic and aircraft landings and takeoffs.

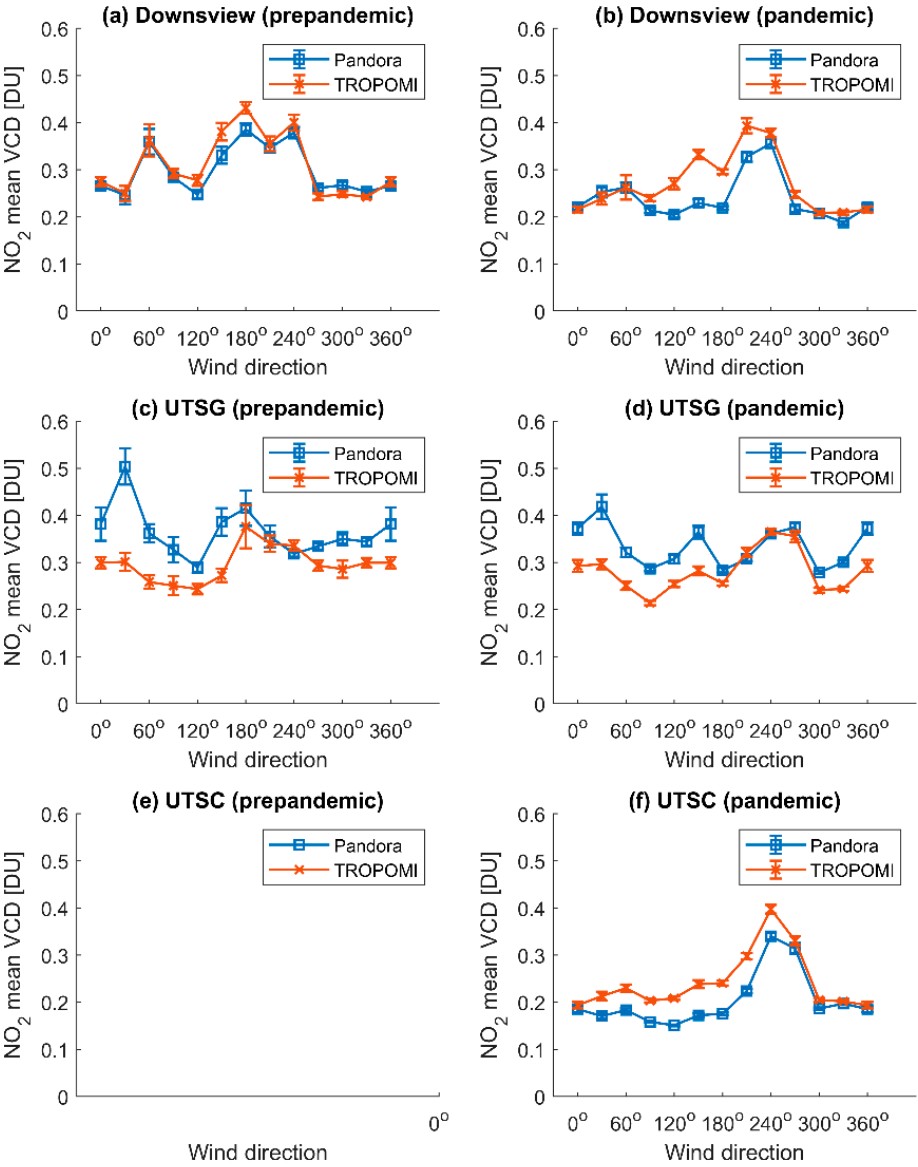

**Figure 11.** TROPOMI and Pandora NO₂ total column (VCD) measurements at Downsview, UTSG, and UTSC binned by wind direction, using data from before the pandemic (**a**,**c**,**e**) and since pandemic (**b**,**d**,**f**). Error bars are the standard error of the mean. Unlike Figures 4 and 6, only Pandora measurements taken within ±1 h around the TROPOMI overpass time were used (observations from 15 March to 15 September for each year).

The relatively larger discrepancy at 30° wind directions at UTSG was due to much less coincident observations from these wind directions (i.e., consistent with previous findings in Zhao et al. [18]). More importantly, Figure 11 shows the regional emissions changed during these two observation periods. In general, both Pandora and TROPOMI show a generally decreased $NO_2$ from most of the wind directions. Higher $NO_2$ columns are found from the north of the UTSG site during the pandemic by Pandora, while this feature is captured by TROPOMI but not as strong as Pandora's observations. We should note that the $NO_2$ wind-direction distribution patterns shown in Figures 4a, 6a,c and 11 are not directly comparable. For Figure 11, only near-local noon data were used (i.e., satellite overpass time), while other figures were produced with entire observational hours. For example, as shown in Figures 4 and 7, the $NO_2$ column and surface values could have strong diurnal variations due to photo-chemistry and boundary layer dynamics.

## 6. Conclusions

Five Canadian Pandora sites have been established in the Greater Toronto Area (GTA) to measure the total columns of atmospheric trace gases such as $NO_2$. Most of these sites are co-located or close to NAPS surface air quality monitoring stations. In this work, we presented remote sensing and in situ $NO_2$ observations before and since the COVID-19 pandemic from three of the five Pandora sites in the GTA. The analysis results show that the $NO_2$ reduction during the pandemic was not homogeneous in space or in time. During the pandemic, $NO_2$ concentrations showed a greater decline in air masses from the south and south-west (downtown and airport areas of the city) than from other directions. More importantly, the measurements reflected the changes in local traffic patterns due to the pandemic, e.g., the rush hour $NO_2$ emission peak vanished at the Downsview site in 2020 and 2021. The overall $NO_2$ emissions during the weekdays was almost similar to emissions during the weekends of the pre-pandemic period.

Both surface and Pandora column $NO_2$ observations demonstrate a substantial change in $NO_2$ values during the COVID-19 related shutdown. For the period from 15 March to 15 September, the tropospheric $NO_2$ values over Downsview were outside the 1-sigma envelope. On average, the tropospheric $NO_2$ values were 26% and 23% below the pre-pandemic averages for weekdays and weekends, respectively. However, the $NO_2$ reduction was not homogeneous spatially or temporally. The observations declined up to 40% from 180° wind directions (from downtown). Temporally, the reduction was up to 43% at 10 am.

Although there is no doubt that the $NO_2$ values showed a substantial decline during the early stage of the pandemic (i.e., April to September 2020), part of this decline can be attributed to the long-term decreasing trend. For the Toronto East site, on top of the significant decreasing trend of surface $NO_2$ ($-0.58 \pm 0.04$ ppbv/yr; $-3.4\%$/yr), the observations from April to September 2020 showed a significant deviation ($-1.53 \pm 1.29$ ppbv) from pre-pandemic conditions on 1-sigma level. Similar analyses were completed for the Pandora observation records. All results revealed that, from April to September 2020, $NO_2$ emissions in the City of Toronto were significantly different from normal conditions, even if we consider the $NO_2$ trends.

The new wind-based satellite validation technique is used to examine the TROPOMI performance during the pandemic. The results show the satellite also successfully captured and revealed the regional air quality changes similar to ground-based instruments. However, due to the limited temporal resolution of TROPOMI, some critical monitoring information is still missing (e.g., diurnal variation) and this gap will be filled by upcoming high-resolution gestational satellite data products (e.g., TEMPO [22]).

**Author Contributions:** Conceptualization, X.Z. and V.F.; methodology, X.Z. and V.F.; software, A.C., M.T. and C.M.; validation, X.Z., Y.S., D.G. and H.E.; formal analysis, X.Z. and R.A.; investigation, X.Z. and R.A.; resources, V.F., S.C.L. and J.D.; data curation, J.D., D.W., K.S., R.A., A.O., R.S. and I.A.; writing—original draft preparation, X.Z.; writing—review and editing, V.F. and all co-authors; visualization, X.Z.; project administration, S.C.L. and T.H.; funding acquisition, S.C.L. and V.F. All authors have read and agreed to the published version of the manuscript.

**Funding:** This research received no external funding.

**Data Availability Statement:** Pandora observations are available via PGN (http://data.pandonia-global-network.org/, last accessed on 21 March 2022). NAPS in situ data can be downloaded from https://www.canada.ca/en/environment-climate-change/services/air-pollution/monitoring-networks-data/national-air-pollution-program.html, last accessed on 21 March 2022.TROPOMI data can be downloaded from https://s5phub.copernicus.eu, last accessed on 21 March 2022. The TROPOMI ECCC research product is available at http://collaboration.cmc.ec.gc.ca/cmc/arqi/, last accessed on 21 March 2022.

**Acknowledgments:** We thank Orfeo Colebatch from the University of Toronto, Kristof Bognar from the University of Saskatoon (formerly at the University of Toronto), Daniel Santana Diaz from PGN, and Michael Gray from NASA for their technical support of Canadian Pandora measurements. We thank the staff at the Ontario Ministry of the Environment, Conservation and Parks who maintained the NAPS ambient air quality monitoring stations in Toronto. We acknowledge the NASA Earth Science Division for providing OMI NO2 SPv3.1 data. The Sentinel 5 Precursor TROPOMI Level 2 product was developed with funding from the Netherlands Space Office (NSO) and processed with funding from the European Space Agency (ESA). X.Z. thanks David Tarasick and Anne Marie Macdonald from ECCC for discussions of $NO_2$ seasonal trends and surface observation results. The Canadian Pandora measurement program had great support from Nader Abuhassan and Robert Swap from NASA in the past years. The PGN is a bilateral project supported with funding from NASA and ESA.

**Conflicts of Interest:** The authors declare no conflict of interest.

## Appendix A

The surface $NO_2$ observations from the NAPS Toronto North site show a significant decreasing trend in Figure A1. From the time series, the surface $NO_2$ observations look continuous. Significant decreasing trends of $-3.8\%$/yr and $-3.7\%$/yr are found with and without including observations during the early stage of the pandemic (i.e., April to September 2020), respectively. However, the observations might not truly reflect the change in local emissions due to the site relocation in 2017, as shown in Figure A2. After relocating the site (only 4 km to the west of the old site), the wind directions where high $NO_2$ concentrations used to be observed, show clear changes. It is also worth pointing out that, following the analysis described in Section 4.2, if we ignore the site relocation, the surface $NO_2$ values at the Toronto North site from April to September 2020 only show a non-significant deviation with considering the overall $NO_2$ decline trend on the 1 sigma level ($-0.93 \pm 1.33$ ppbv).

Similar analyses were made for Pandora's observations at this site. However, the record is only seven years (2015–2021), covering a shorter period with less clear changing of seasonal amplitude. Thus, the trend fitting model is simplified to one with a fixed seasonal amplitude (similar to Equation (2)), but without the local regression function). The fitted results are shown in Figure A3. This result also confirmed that the $NO_2$ tropospheric column observations in the early stage of the pandemic were significantly low ($-0.04$ DU), even considering the overall $NO_2$ decreasing trend in this area.

The Google Community Mobility Reports show that the visits to transit stations (e.g., subway, bus, and train stations) and workplaces in Toronto decreased by 73% and 67% in early April 2020 (see Figure A4). These local traffic and commuter pattern changes would affect road $NO_2$ emissions.

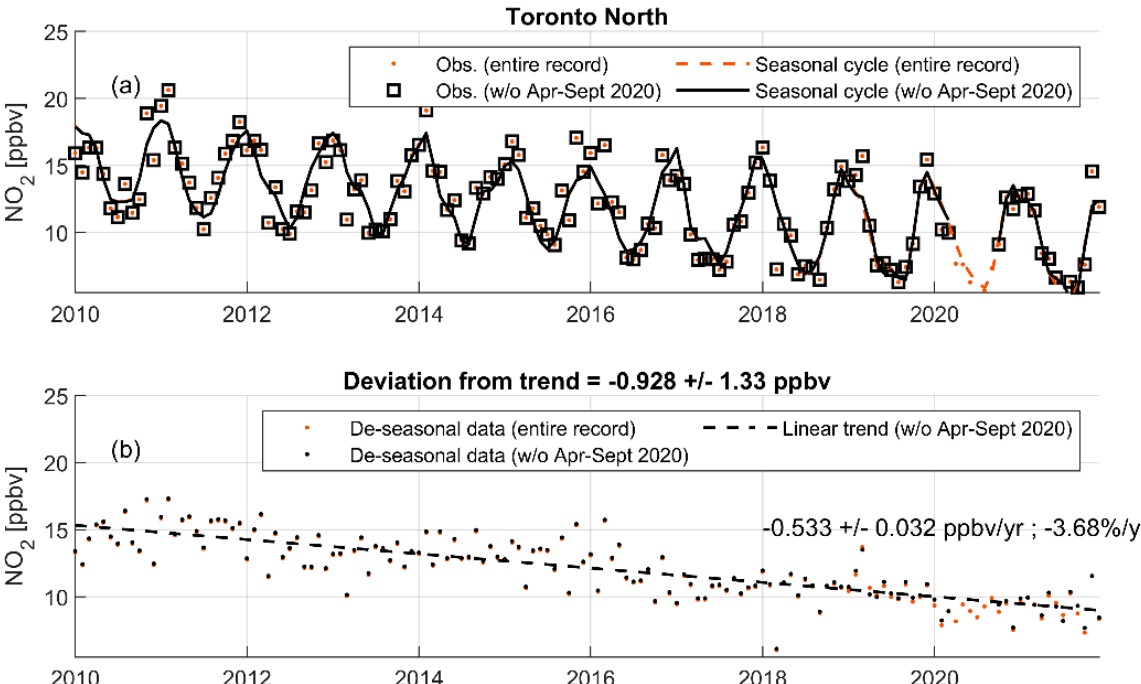

**Figure A1.** Surface NO$_2$ trend fitting results from the statistical model at the Toronto North site. Red symbols represent data and fittings for the entire observation record. Black symbols represent data and fittings for the records without data from April to September 2020. Panel (**a**) shows the observations and fitted seasonal cycles; panel (**b**) shows the de-seasonal data and fitted linear trend.

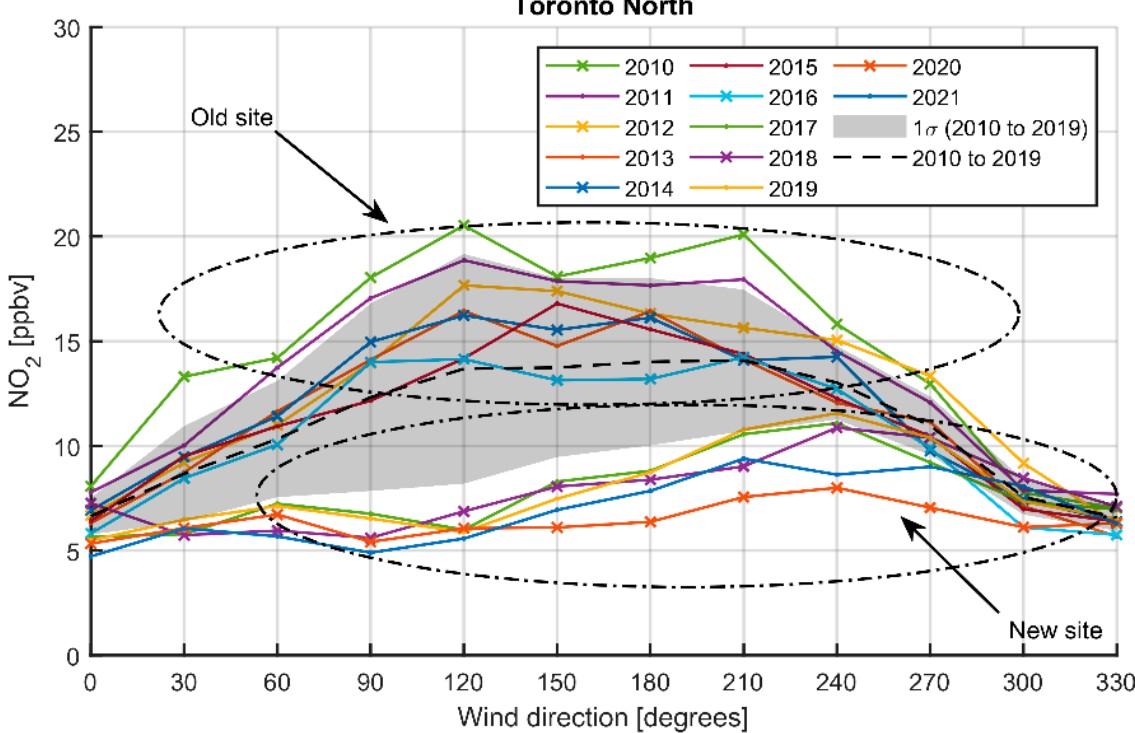

**Figure A2.** In situ measured NO$_2$ surface concentration at Toronto North site binned by local standard time (LST) in hours and wind directions (observations from 15 March to 15 September for each year). The black dashed line is the mean of observations from pre-pandemic with shading area representing the standard deviation of the mean.

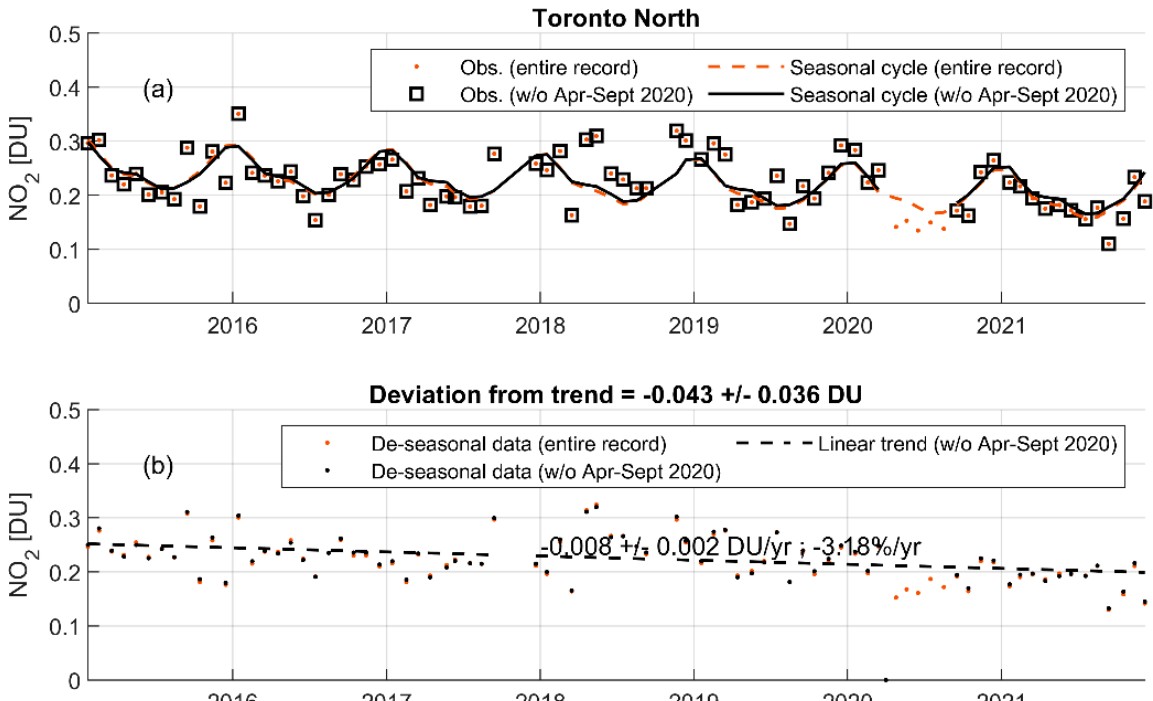

**Figure A3.** Pandora $NO_2$ tropospheric column fitting results from the statistical model at Downsview (i.e., NAPS Toronto North site). Red symbols represent data and fittings for the entire observation record. Black symbols represent data and fittings for the records without data from April to September 2020. Panel (**a**) shows the observations and fitted seasonal cycles; panel (**b**) shows the de-seasonal data and fitted linear trend.

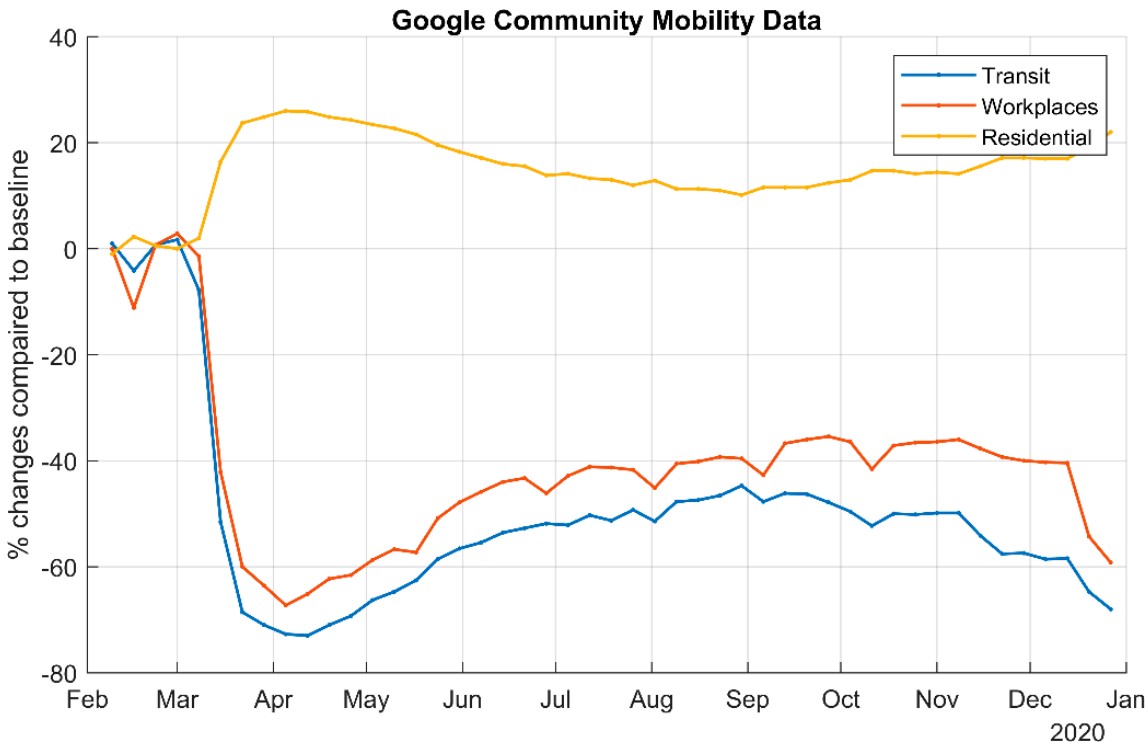

**Figure A4.** Weekly averaged percentage mobility changes in Toronto (2020). Data from Google Community Mobility Report (© Google).

**Appendix B**

The standard TROPOMI NO$_2$ data product [18] produced by KNMI was also evaluated in the same fashion with Pandora observations. Consistent with previous findings, the results showed both ECCC and S5P-PAL data products can tack the horizontal distribution patterns of the NO$_2$ pollutants. Figures A5–A8 show the regression, histogram, and wind-based analysis results. In general, similar to ECCC recalculated NO$_2$ data product, the agreement between satellite and ground-based observations was improved during the pandemic (i.e., lower bias and better correlation coefficient). Figures A6 and A7 show that, as expected, an increased (more realistic) column over the city center due to a higher-resolution a priori in ECCC data, but otherwise the columns in S5P-PAL have increased (before July 2021) compared to previous versions (1.2/1.3 and even 1.4).

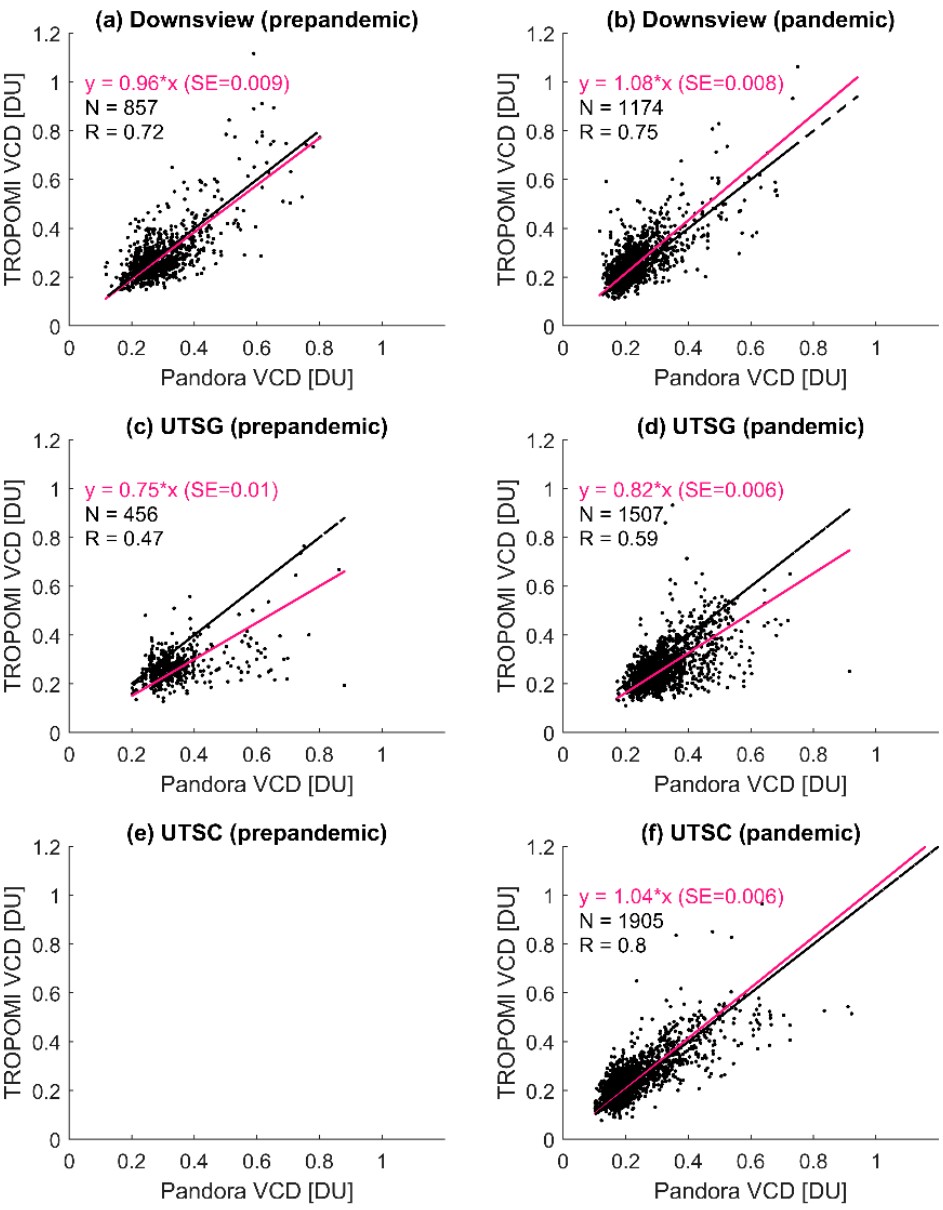

**Figure A5.** TROPOMI (S5P-PAL standard product) vs. Pandora NO$_2$ total column (VCD) measurements at Downsview, UTSG, and UTSC, using data from before the pandemic (**a**,**c**,**e**) and since pandemic (**b**,**d**,**f**). The magenta line is the linear fit with intercept set to 0, and the black line is the one-to-one line. Pandora measurements taken within ±1 h around the TROPOMI overpass time were used (observations from 15 March to 15 September for each year).

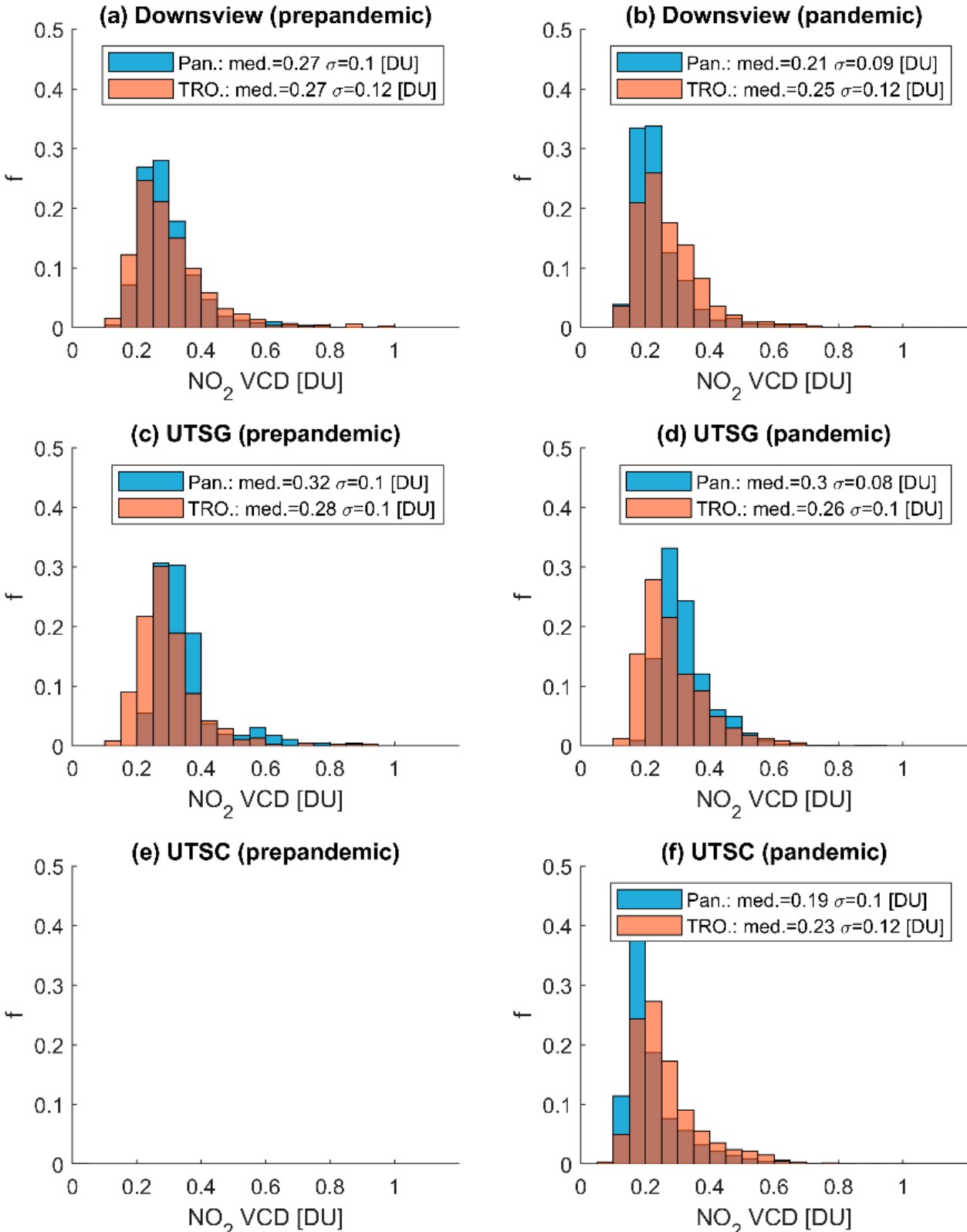

**Figure A6.** Normalized histogram of TROPOMI (ECCC product) and Pandora NO$_2$ total column (VCD) measurements at Downsview, UTSG, and UTSC, using data from before the pandemic (**a,c,e**) and since pandemic (**b,d,f**). The median and standard deviation of observations are shown in the legends. Pandora measurements taken within ±1 h around the TROPOMI overpass time were used (observations from 15 March to 15 September for each year).

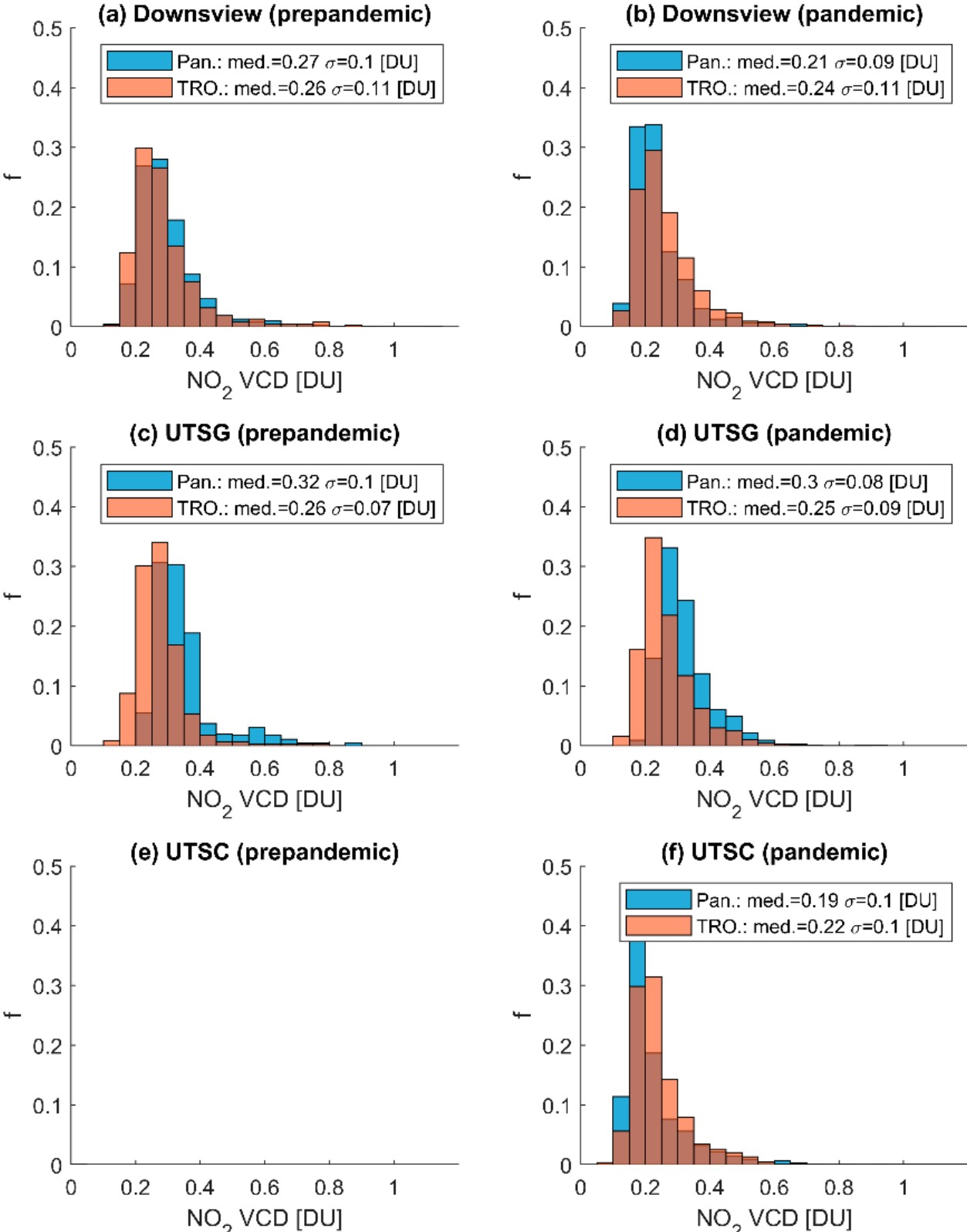

**Figure A7.** Normalized histogram of TROPOMI (S5P-PAL standard product) and Pandora NO$_2$ total column (VCD) measurements at Downsview, UTSG, and UTSC, using data from before the pandemic (**a**,**c**,**e**) and since pandemic (**b**,**d**,**f**). The median and standard deviation of observations are shown in the legends. Pandora measurements taken within $\pm1$ h around the TROPOMI overpass time were used (observations from 15 March to 15 September for each year).

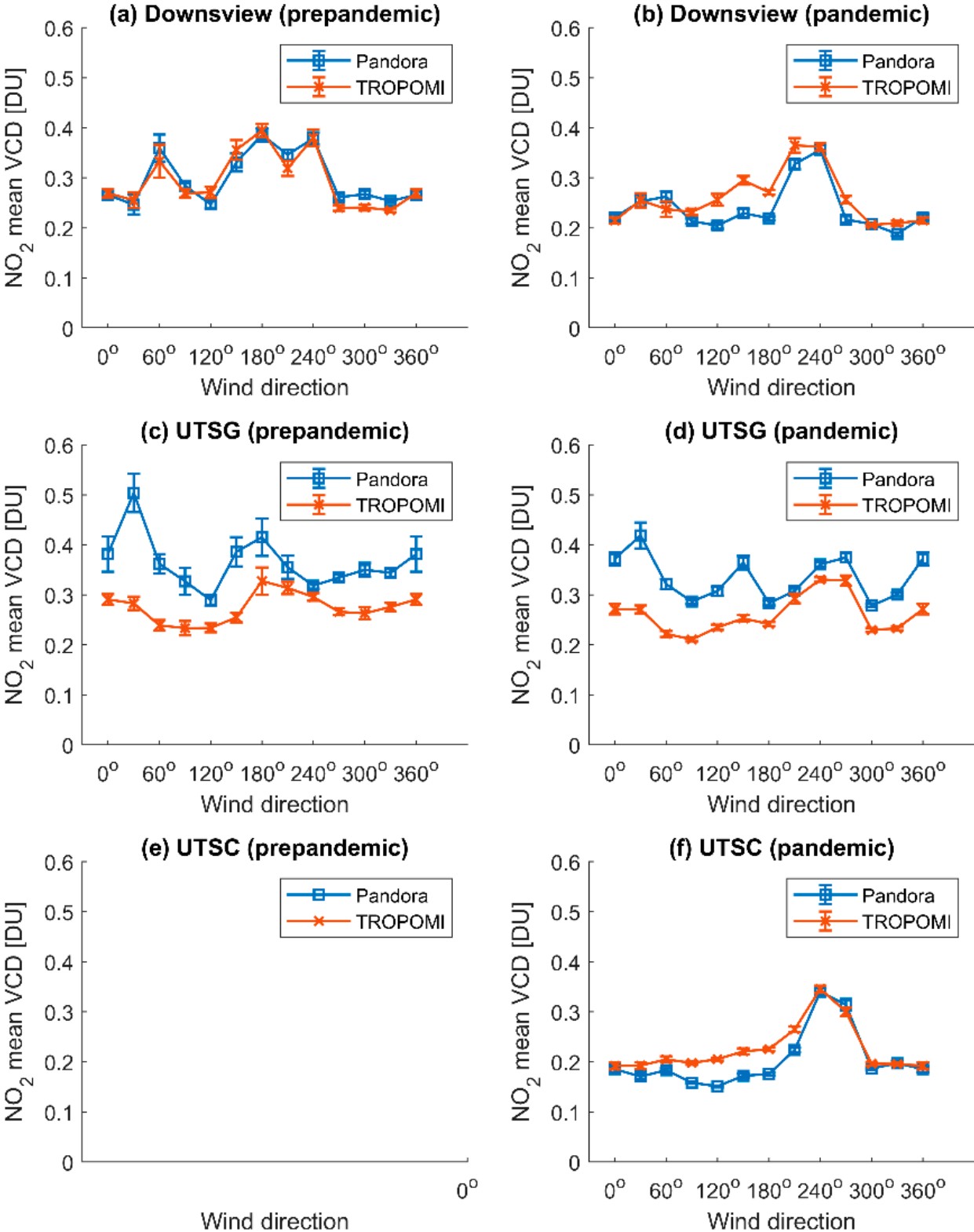

**Figure A8.** TROPOMI (S5P-PAL standard product) and Pandora NO$_2$ total column (VCD) measurements at Downsview, UTSG, and UTSC binned by wind direction, using data from before the pandemic (**a**,**c**,**e**) and since pandemic (**b**,**d**,**f**). Error bars are the standard error of the mean. Unlike Figures 4 and 6, only Pandora measurements taken within $\pm 1$ h around the TROPOMI overpass time were used (observations from 15 March to 15 September for each year).

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
