# Peer review of "Tropospheric and Surface Nitrogen Dioxide Changes in the Greater Toronto Area during the First Two Years of the COVID-19 Pandemic"

_remotesensing, doi:10.3390/rs14071625_

Round 1
Reviewer 1 Report
This is a very well written paper focusing on tropospheric and surface NO2 changes over different areas around Toronto the two years of the COVID-19 pandemic. The paper definitely merits to be published after minor changes have been incorporated into the introduction which is not sufficient. More specifically, the authors should extend their introduction and do a connection with other global and regional studies focusing on NO2 level drop during the pandemic. How big was the decrease compared to other areas around the world? There are numerous studies from China (e.g. Feng et al., 2020; Liu et al., 2020; Liu et al., 2021) to Europe (e.g. Barré et al., 2021; Akritidis et al., 2021), the US (e.g. Qu et al., 2021) and global studies (e.g. Bauwens et al., 2020; Keller et al., 2021; Cooper et al., 2022) focusing on the effect of COVID-19 restrictions on NO2 levels. The authors are urged to discuss their results in accordance with other studies focusing on other locations and cite the suggested papers and other related papers.
Akritidis, D.; Zanis, P.; Georgoulias, A.K.; Papakosta, E.; Tzoumaka, P.; Kelessis, A. Implications of COVID-19 Restriction Measures in Urban Air Quality of Thessaloniki, Greece: A Machine Learning Approach. Atmosphere 2021, 12, 1500. https://doi.org/10.3390/atmos12111500
Barré, J., et al.: Estimating lockdown-induced European NO2 changes using satellite and surface observations and air quality models, Atmos. Chem. Phys., 21, 7373–7394, https://doi.org/10.5194/acp-21-7373-2021, 2021.
Bauwens, M., Compernolle, S., Stavrakou, T., Müller, J.-F., van Gent, J., Eskes, H., et al. (2020). Impact of coronavirus outbreak on NO2 pollution assessed using TROPOMI and OMI observations. Geophysical Research Letters, 47, e2020GL087978. https://doi.org/10.1029/2020GL087978.
Cooper, M.J., Martin, R.V., Hammer, M.S. et al. Global fine-scale changes in ambient NO2 during COVID-19 lockdowns. Nature 601, 380–387 (2022). https://doi.org/10.1038/s41586-021-04229-0.
Feng, S., Jiang, F., Wang, H., Wang, H., Ju, W., Shen, Y., et al. (2020). NOx emission changes over China during the COVID-19 epidemic inferred from surface NO2 observations. Geophysical Research Letters, 47, e2020GL090080. https://doi.org/10.1029/2020GL090080.
Keller, C. A., Evans, M. J., Knowland, K. E., Hasenkopf, C. A., Modekurty, S., Lucchesi, R. A., Oda, T., Franca, B. B., Mandarino, F. C., Díaz Suárez, M. V., Ryan, R. G., Fakes, L. H., and Pawson, S.: Global impact of COVID-19 restrictions on the surface concentrations of nitrogen dioxide and ozone, Atmos. Chem. Phys., 21, 3555–3592, https://doi.org/10.5194/acp-21-3555-2021, 2021.
Liu F., et al. ., Abrupt decline in tropospheric nitrogen dioxide over China after the outbreak of COVID-19. Sci. Adv. 6, eabc2992 (2020).
Liu, S., Valks, P., Beirle, S. et al. Nitrogen dioxide decline and rebound observed by GOME-2 and TROPOMI during COVID-19 pandemic. Air Qual Atmos Health 14, 1737–1755 (2021). https://doi.org/10.1007/s11869-021-01046-2.
Qu, Z., Jacob, D. J., Silvern, R. F., Shah, V., Campbell, P. C., Valin, L. C., & Murray, L. T. (2021). US COVID-19 shutdown demonstrates importance of background NO2 in inferring NOx emissions from satellite NO2 observations. Geophysical Research Letters, 48, e2021GL092783. https://doi.org/10.1029/2021GL092783.
Author Response
The new/revised text in the modified manuscript is given in red (italicized).
Reply to Referee #1:
We thank the referee for these important suggestions and we appreciate the comments. There have been lots of good and detailed research on COVID-19 NO2 emission changes since 2020. We have followed the suggestion and included a paragraph to provide readers with some general background on these research works.
Since 2020, many research studies have been carried out to evaluate the NO2 pollution changes during the early period or first year of the COVID-19 pandemic. Numerous studies were done regionally (e.g., [25–31]) and globally (e.g., [32–34]), with various observations methods. For example, Bauwens et al. [32] reported the average NO2 column, observed by TROPOMI, dropped by 40%, 38%, and 20%, over Chinese, American, and European cities, respectively by April 2020 compared to pre-COVID pandemic measurements. Cooper et al. [33] quantified NO2 changes in more than 200 cities worldwide and reported mean surface NO2 concentrations are 29% ± 3% lower in countries with strict lockdown conditions than in those without in 2020. To our best knowledge, this paper is the first peer-review study of the COVID-19 pandemic NO2 changes that cover the period of not only 2020 but also 2021.
- Barré, J.; Petetin, H.; Colette, A.; Guevara, M.; Peuch, V.-H.; Rouil, L.; Engelen, R.; Inness, A.; Flemming, J.; Pérez García-Pando, C.; et al. Estimating Lockdown-Induced European NO2 Changes Using Satellite and Surface Observations and Air Quality Models. Atmos. Chem. Phys. 2021, 21, 7373–7394, doi:10.5194/acp-21-7373-2021.
- Akritidis, D.; Zanis, P.; Georgoulias, A.K.; Papakosta, E.; Tzoumaka, P.; Kelessis, A. Implications of COVID-19 Restriction Measures in Urban Air Quality of Thessaloniki, Greece: A Machine Learning Approach. Atmosphere 2021, 12, 1500, doi:10.3390/atmos12111500.
- Badia, A.; Langemeyer, J.; Codina, X.; Gilabert, J.; Guilera, N.; Vidal, V.; Segura, R.; Vives, M.; Villalba, G. A Take-Home Message from COVID-19 on Urban Air Pollution Reduction through Mobility Limitations and Teleworking. npj Urban Sustain 2021, 1, 1–10, doi:10.1038/s42949-021-00037-7.
- Feng, S.; Jiang, F.; Wang, H.; Wang, H.; Ju, W.; Shen, Y.; Zheng, Y.; Wu, Z.; Ding, A. NOx Emission Changes Over China During the COVID-19 Epidemic Inferred From Surface NO2 Observations. Geophys. Res. Lett. 2020, 47, e2020GL090080, doi:10.1029/2020GL090080.
- Li, J.; Tartarini, F. Changes in Air Quality during the COVID-19 Lockdown in Singapore and Associations with Human Mobility Trends. Aerosol Air Qual. Res. 2020, 20, 1748–1758, doi:10.4209/aaqr.2020.06.0303.
- Liu, S.; Valks, P.; Beirle, S.; Loyola, D.G. Nitrogen Dioxide Decline and Rebound Observed by GOME-2 and TROPOMI during COVID-19 Pandemic. Air Qual Atmos Health 2021, 14, 1737–1755, doi:10.1007/s11869-021-01046-2.
- Qu, Z.; Jacob, D.J.; Silvern, R.F.; Shah, V.; Campbell, P.C.; Valin, L.C.; Murray, L.T. US COVID-19 Shutdown Demonstrates Importance of Background NO2 in Inferring NOx Emissions From Satellite NO2 Observations. Geophys. Res. Lett. 2021, 48, e2021GL092783, doi:10.1029/2021GL092783.
- Bauwens, M.; Compernolle, S.; Stavrakou, T.; Müller, J.-F.; van Gent, J.; Eskes, H.; Levelt, P.F.; van der A, R.; Veefkind, J.P.; Vlietinck, J.; et al. Impact of Coronavirus Outbreak on NO2 Pollution Assessed Using TROPOMI and OMI Observations. Geophys. Res. Lett. 2020, 47, e2020GL087978, doi:10.1029/2020GL087978.
- Cooper, M.J.; Martin, R.V.; Hammer, M.S.; Levelt, P.F.; Veefkind, P.; Lamsal, L.N.; Krotkov, N.A.; Brook, J.R.; McLinden, C.A. Global Fine-Scale Changes in Ambient NO2 during COVID-19 Lockdowns. Nature 2022, 601, 380–387, doi:10.1038/s41586-021-04229-0.
- Keller, C.A.; Evans, M.J.; Knowland, K.E.; Hasenkopf, C.A.; Modekurty, S.; Lucchesi, R.A.; Oda, T.; Franca, B.B.; Mandarino, F.C.; Díaz Suárez, M.V.; et al. Global Impact of COVID-19 Restrictions on the Surface Concentrations of Nitrogen Dioxide and Ozone. Atmos. Chem. Phys. 2021, 21, 3555–3592, doi:10.5194/acp-21-3555-2021.
Reviewer 2 Report
The paper shows a relatively comprehensive analysis of the tropospheric and surface NO2 across years. The paper is structured well with sufficient data. However, there are still some issues to be solved before it is publishable.
Major comments:
1. The authors repetitively claimed the association between local traffic (or, commuter) and NO2 but with no data to support it. I would recommend authors to find available sources of the local traffic load. Two available options could be the database published by Google and Apple. Another one could be the reported car park availability if it is applicable in your city. Please find the paper below for details of the sources of mobility data and discuss with it.
Li, J., & Tartarini, F. (2020). Changes in Air Quality during the COVID-19 Lockdown in Singapore and Associations with Human Mobility Trends. Aerosol and Air Quality Research, 20(8), 1748–1758. https://doi.org/10.4209/aaqr.2020.06.0303
Badia, A., Langemeyer, J., Codina, X., Gilabert, J., Guilera, N., Vidal, V., Segura, R., Vives, M., & Villalba, G. (2021). A take-home message from COVID-19 on urban air pollution reduction through mobility limitations and teleworking. Npj Urban Sustainability, 1(1), 1–10. https://doi.org/10.1038/s42949-021-00037-7
2. If the surface NO2 decreases yearly as you mentioned in Figure 8, your reported decline of NO2 caused by the pandemic can be overestimated because it declines every year intrinsically with and without the pandemic.
Minor:
1. Line 39-40. Please remove the numbers after the keywords.
2. Please remove the left borderlines for Figures 6 and 7.
Author Response
The new/revised text in the modified manuscript is given in red (italicized).
Reply to Referee #2:
We thank the referee for these important suggestions and we appreciate the comments. We analyzed Google and Apple mobility data as suggested. We included a paragraph and a new figure in Appendix A (Google data) to show some local mobility data to support the discussions. The suggested references were also provided in the general discussion part of the Introduction (see our reply to referee #1).
The Google Community Mobility Reports show the visits to transit stations (e.g., subway, bus, and train stations) and workplaces in Toronto decreased by 73% and 67% in early April 2020 (see Fig. A4). These local traffic and commuter pattern changes would affect road NO2 emissions.
(see Figure A4 in revised paper)
Figure A4. Weekly averaged percentage mobility changes in Toronto (2020). Data from Google Community Mobility Report (© Google).
We fully agree with the referee that the reported NO2 decline is partially due to this long-term NO2 reduction effort in the GTA (this was included in the paper). We included the detailed trend analysis in Section 4.2 and Appendix A to evaluate if the reduction due to the COVID-19 pandemic is statistically significant, when considering the long-term decreasing trend.
On the other hand, one of the major findings of this work is the inhomogeneity of the NO2 reduction during the pandemic. The overall surface NO2 trend only reflects conditions from all wind directions and all-time (i.e., 24 hr). Whereas, we demonstrated that the NO2 decline is stronger from more polluted city centre areas and during morning rush hours periods. We found the COVID-19 pandemic caused these types of NO2 declines significantly, but their signals might be overlooked by “simple” long-term trend analysis. However, from the current results, we were able to conclude that even considering the overall NO2 decreasing trend in this area, the surface NO2 reduction in this early pandemic period is still significant on more than the 2-sigma level (see the detailed discussions in Section 4.2).
The suggested minor corrections have been implemented.
Reviewer 3 Report
This is a good research paper. The authors analyzed the variation characteristics of Tropospheric and Surface Nitrogen Dioxide Changes in the Greater Toronto Area using 7 years of various types of data. The data is very rich and the results are reliable. The text expression is also accurate. This is a good article to study the impact of COVID Pandemic. I personally feel that it has met the publication requirements of the journal. It is recommended to accept it for publication.
Author Response
Reply to Referee #3:
We thank the referee for the time to review the paper and we appreciate the comments.
Reviewer 4 Report
I found the paper well written and interesting.
I only a minor comment: if possible, I would ask the authors to be a bit more specific about the spatial resolution of the used satellite data (both original and after processing), its temporal resolution and the uncertainties associated to it.
Author Response
The new/revised text in the modified manuscript is given in red (italicized).
Reply to Referee #4:
We thank the referee for the time to review the paper and we appreciate the suggestion. The overall uncertainty of TROPOMI NO2 data is 0.032 DU (was included in Section 2.3). We also included a short paragraph in Section 2.3 to give more details about its temporal resolution and the data uncertainties.
Detailed uncertainties estimations of TROPOMI NO2 data products can be found in Zhao et al. [18]. As a sun-synchronous satellite instrument, TROPOMI only measures once per day over most mid-latitude regions (including the GTA). At nadir, TROPOMI pixel sizes were 3.5 × 7 km2 at the beginning of operation and were reduced to 3.5 × 5.6 km2 on 6 August 2019.